

# Composition and volatility of SOA formed from oxidation of real tree emissions compared to single VOC-systems

Arttu Ylisirniö[1], Angela Buchholz[1], Claudia Mohr [2,3], Zijun Li[1], Luis Barreira[1,4], Andrew Lambe[5], Celia Faiola[1,6], Eetu Kari[1,a], Taina Yli-Juuti[1], Sergey A. Nizkorodov[7], Douglas R. Worsnop[5], Annele Virtanen[1], Siegfried Schobesberger[1]

[1]Department of Applied Physics, University of Eastern Finland, Kuopio, Finland
[2]Institute of Meteorology and Climate Research, Karlsruhe Institute of Technology, Karlsruhe, Germany
[3]Department of Environmental Science and Analytical Chemistry, Stockholm University, Stockholm, Sweden
[4]Atmospheric Composition Research, Finnish Meteorological Institute, Helsinki, Finland
[5]Center for Aerosol and Cloud Chemistry, Aerodyne Research, Inc., Billerica, MA, USA
[6]Department of Ecology and Evolutionary Biology, University of California, Irvine, Irvine, USA
[7]Department of Chemistry, University of California, Irvine, Irvine, USA
[a]currently at: Neste Oyj, Espoo, Finland

*Correspondence to*: Arttu Ylisirniö (arttu.ylisirnio@uef.fi)

**Abstract.** Secondary organic aerosol (SOA) is an important constituent of the atmosphere where SOA particles are formed chiefly by the condensation or reactive uptake of oxidation products of volatile organic compounds (VOC). The mass yield in SOA particle formation, as well as the chemical composition and volatility of the particles are determined by the identity of the VOC precursor(s) and the oxidation conditions they experience. In this study, we used an oxidation flow reactor to generate biogenic SOA from the oxidation of Scots pine emissions. Mass yields, chemical composition, and volatility of the SOA particles were characterized and compared with SOA particles formed from oxidation of α-pinene and of a mixture of acyclic/monocyclic sesquiterpenes (farnesenes and bisabolenes), which are significant components of the Scots pine emissions. SOA mass yields for Scots pine emissions dominated by farnesenes were lower than for α-pinene, but higher than for the artificial mixture of farnesenes and bisabolenes. The reduction in the SOA yield in the farnesenes and bisabolenes dominated mixtures is due to C=C bond scission in these acyclic/monocyclic sesquiterpenes during ozonolysis leading to smaller and generally more volatile products. SOA particles from the oxidation of Scots pine emission had similar or lower volatility than SOA particles formed from either of single precursor. Applying physical stress to the Scots pine plants increased monoterpene emissions, which further decreased SOA particle volatility and increased SOA mass yield. Our results highlight the need to account for the chemical complexity and structure of real-world biogenic VOC emissions and stress-induced changes to plant emissions when modelling SOA production and properties in the atmosphere. These results emphasize that simple increase or decrease of relative monoterpene and sesquiterpene emissions should not be used as indicator of SOA particle volatility.



# 1 Introduction

Secondary organic aerosol (SOA) formed from oxidation of volatile organic compounds (VOCs) contributes a large fraction
to the total aerosol mass in the boreal forests of the northern hemisphere (Hallquist et al., 2009; Jimenez et al., 2009;
Riipinen et al., 2012). The chemical transformation of primary VOC emissions to SOA particles, which have an important
climate impact (Hallquist et al., 2009), is a complicated cascade of gas-phase oxidation and multi-phase aging reactions. The
physical properties of SOA are dictated by the chemical complexity of the initial VOC emissions and the oxidative
conditions they experience (Glasius and Goldstein, 2016).


The formation and growth of SOA particles is often described by the absorptive partitioning of organic vapours (e.g.
terpenoid oxidation products) between gas and particle phase (Donahue et al., 2011; Pankow, 1994). The main property
determining how readily organic molecules enter and stay in the particle phase is their volatility, usually expressed as
saturation vapor pressure ($Psat$) or saturation mass concentration ($C^*$) in air. The volatility of a specific compounds in turn
determined by both its molar mass and functional group composition (Capouet and Müller, 2006; Pankow and Asher, 2008).
Oxidation of a single VOC precursor produces a wide variety of semi- and low-volatility compounds, which are able to
condense into the particle phase according to their $C^*$. The uptake of oxidation products may also involve or be facilitated by
heterogeneous reactions.

In boreal forest environments, VOC emissions are dominated by monoterpenes. As the globally most important monoterpene
(Spanke et al., 2001), α-pinene has hence been most commonly used as a model species in laboratory SOA studies.
Consequently, a-pinene has been used as a proxy for all other monoterpenes in atmospheric models using its properties to
describe the atmospheric oxidation and contribution to SOA particles of all other monoterpenes (Holopainen et al., 2017).

However, the VOC emission patterns of vegetation vary significantly depending on locations, environmental conditions and
even genotypes of each plant. Moreover, differences in emissions can be even found between two plants of the same species
with same age located in the same environment (Bäck et al., 2012; Hakola et al., 2017; Holopainen and Gershenzon, 2010).
Even though the VOC emissions from boreal forests are thought to mostly consist of monoterpenes, Hellén et al., (2018)
recently showed a significant contribution of sesquiterpenes to the total VOC budget in boreal forests. This finding is in line
with an earlier study by Hakola et al. (2017) that showed high emissions of sesquiterpenes measured from branch enclosures
in a coniferous forest in Finland during early spring and late autumn. Mixtures of different VOCs also influence the oxidant
reactivity and oxidant product distribution compared to single VOC precursor oxidation (Mcfiggans et al., 2019), which
highlights the need to use real emissions as precursors while exploring the physical and chemical properties of SOA
particles.






In this study, we present results from a state-of-the-art suite of instruments used to investigate the VOC emission profile of Scots pine saplings (Pinus sylvestris) and subsequent SOA formation from these VOCs in an oxidation flow reactor (OFR). The central instrument in this work is a Filter Inlet for Gases and AEROsols, coupled to a high-resolution time-of-flight Chemical Ionization Mass Spectrometer (FIGAERO-CIMS; Aerodyne Inc. and Tofwerk AG; Lopez-Hilfiker et al., 2014),

which allowed us to characterize the composition and thermal desorption behavior of the SOA particles. For comparison, we performed the same measurements for SOA formed from α-pinene in the same OFR. Prompted by a strong contribution of farnesenes to the emissions from our plants, we also examined SOA formation from a mixture of acyclic/monocyclic sesquiterpenes with major contribution from isomeric farnesenes and bisabolenes. This is the first volatility measurement of SOA particles formed from the oxidation of farnesene and bisabolene since earlier studies involving these sesquiterpenes

have focused on the gas-phase chemistry (Kim et al., 2011; Kourtchev et al., 2009, 2012) or the chemical composition of the SOA particles (Jaoui et al., 2017). Our results show surprising impacts of the VOC mixture on the SOA particle volatility compared to SOA particles formed from a single VOC compound.

## 2 Materials and methods

### 2.1. VOC measurements and SOA production

We conducted experiments with SOA generated from the photo-oxidation of VOCs by hydroxyl radicals (OH) in a Potential Aerosol Mass (PAM) OFR (Kang et al., 2007; Lambe et al., 2011) in the absence of seed particles. In all experiments the operation temperature was 25 ˚C and relative humidity (RH%) was between 40% to 60%. Wavelength of the used UV-lights was 254 nm. The integrated OH exposure in the OFR ranged from approx. $6.6\times10^{10}$ to $2.5\times10^{12}$ molec cm$^{-3}$ s across all experiments as calculated according to methods described by Peng et al., (2015, 2016). This range of OH exposure

corresponds to 0.5 to 19 equivalent days of atmospheric aging at an OH concentration of $1.5\times10^{6}$ molec cm$^{-3}$ (Palm et al., 2016). Depending on the experiment, VOCs were introduced from (1) a plant enclosure containing a 6-year-old Scots pine sapling, (2) from an external diffusion source (α-pinene (Sigma Aldrich, 98 % purity)), or (3) a dynamic dilution system (mixture of acyclic/monocyclic sesquiterpenes, (Sigma Aldrich), Kari et al., 2018). For the Scots pine experiment 4, the plant was injured by making four 0.5-1 cm$^2$ cuts into the bark of the plant to induce higher VOC emissions. A list of all

experimental conditions is provided in Table 1 and Table 2. A schematic of the experimental setup is shown in Fig. 1.

The VOC mixing ratios entering the OFR were continuously monitored using a proton-transfer-reaction time-of-flight mass spectrometer (PTR-ToF-MS, PTR-TOF 8000, Ionicon Analytik Inc.) directly upstream of the OFR inlet but before the addition of ozone to the system. All reported mixing ratios were corrected for this dilution and represent the conditions at the

inlet of the PAM reactor. In addition, to resolve the mixture of terpenoid emissions emitted by the Scots pine sapling, we collected two cartridge samples (Tenax-TA, MARKES etc.) at the beginning of the Scots pine experiment 1 and at the end of the Scots pine experiment 4 for off-line analysis using a Thermal Desorption Gas Chromatograph Mass Spectrometer (TD-



GC–MS, TD: Perkin Elmer, ATD 400, USA, GC–MS: Hewlett Packard, GC 6890, MSD 5973, USA). PTR-ToF-MS calibration procedure is described in Sect. S2.


The measured emission profiles from the Scots pine showed a strong contribution of α- and β-farnesene (Fig. 2), which are acyclic sesquiterpenes. Therefore, we conducted follow up experiments under similar oxidative conditions using a commercially-available mixture of acyclic/monocyclic sesquiterpenes to investigate the effect of such biogenic, unsaturated, acyclic/monocyclic VOCs on SOA properties. This sesquiterpene mixture consisted of isomers of farnesenes and bisabolene,

both of which are found in the emissions of coniferous trees as well as the emissions of Scots pines in this study (Blande et al., 2009; Holopainen and Gershenzon, 2010). For a detailed description of the mixture see Table S1 and Fig. S1. For those follow-up experiments, a nominally identical OFR was used ("PAM 2"). Because of the different use history of the two OFRs, the corresponding results are presented separately. More detailed information on experimental conditions is found in Tables 1 and 2.


## 2.2. SOA particles characterization

SOA particles were examined with a suite of instruments sampling from the outlet of the OFR: an Aerodyne High-Resolution Time-of-Flight Aerosol Mass Spectrometer (AMS), a Scanning Mobility Particle Sizer (SMPS, TSI Inc, Model 3082 with TSI Model 3775 Condensation Particle Counter) and a FIGAERO-CIMS using the iodide ionization scheme (Lee

et al., 2014).

The FIGAERO-CIMS was used to characterize the volatility and chemical composition of the SOA particles for those organic compounds sensitive to iodide cluster ionization. Briefly, in the ionization region of the instrument, an $I^-$ anion preferably clusters with a neutral molecule $M$ which has oxygen or OH-groups in their structure. The neutral molecule is

then observed as $[M+I]^-$ in the mass spectrometer (Iyer et al., 2017; Lee et al., 2014). In the presence of water, collision of an $[H_2O+I]^-$ anion cluster with $M$ may produce the same result. In some cases, the $[M+I]^-$ cluster breaks apart leading to deprotonation of the neutral molecule, which is then observed as $[M-H]^-$, and possibly to other ion fragments. This phenomenon has also been observed in earlier studies (e.g. Lee et al., 2014). For the further analysis, we assume that deprotonation is the only mechanism of declustering, as it is one known to potentially follow already from relatively soft

collisions. Additional declustering may happen by more energetic collisions in the lower-pressure regions of the mass spectrometer (Passananti et al., 2019).

In the FIGAERO inlet, the aerosol particles are collected on a Teflon filter (Zefluor 2 µm PTFE Membrane filter, Pall Corp.) and then evaporated and carried into the instrument with gradually heated nitrogen flow with a heating rate of 11.6 (K/min).

This results in temperature-dependent ion signals for each observed mass spectrum peak (thermograms) that can be related to



the volatility of the collected organic compounds (Lopez-Hilfiker et al 2014, and Lopez-Hilfiker et al., 2015). In this study, two slightly different FIGAERO inlets were used: one in conjunction with the initial experiments using the "PAM 1" reactor ("FIGAERO 1"), the other when using the "PAM 2" reactor later-on ("FIGAERO 2"). The CIMS themselves were nominally identical, and both of the FIGAERO inlets followed the identical principles and were operated identically. The

differences were in the detailed design of the FIGAERO inlets, e.g., the shape of filter collection tray and exact positioning of temperature sensor. These changes lead to apparent shifts of the measured thermograms. To account for this, we performed instrument specific calibrations for both FIGAERO inlets, which is described in more detail in Sect. 2.2.2.

AMS and SMPS were used to continuously monitor the output SOA particle mass and size distribution from the OFR and to

determine the point when the particle concentrations and distributions had stabilized for a given OFR condition. Then the filter collections for FIGAERO-CIMS was started, so that only steady-state SOA was sampled.

### 2.2.1 FIGAERO-CIMS data-analysis

The FIGAERO-CIMS data was analysed with the MATLAB-based tofTools (Junninen et al., 2010) software, including the identification of elemental compositions (creation of peak lists) based on the high resolution (HR) peak fitting with mass

accuracy of 5 ppm. All data shown here is based on the HR-fitting of the mass spectra using peak lists covering the full spectra. The presented mass values are those of the neutral composition ($M$), derived by subtracting the molecular mass of $I^-$ when $[M+I]^-$ was observed or by adding the molecular mass of $H$ when $[M-H]^-$ was observed. The assigned formulas were constrained to contain only C, H and O elements; any signal peak appearing to contain other elements was considered background and excluded from further analysis. This background consisted mostly of fluorine containing compounds, which

we assume to originate from the FIGAERO inlet manifold or the collection filter, both of which are made of PTFE.

### 2.2.2 T$_{max}$ to saturation concentration calculations

Typically, the thermograms collected from the heat-induced SOA evaporation in the FIGAERO filter comprise a clearly defined peak, i.e., a temperature at which the largest signal is observed ($T_{max}$). It has been shown that $T_{max}$ correlates with the

volatility of the desorbing organic compound, typically expressed as saturation concentration $C^*$, at least for systems with limited number of compounds (Bannan et al., 2018; Lopez-Hilfiker et al., 2014). We determined the relationship between $T_{max}$ and $C^*$ for both FIGAERO inlets independently, based on the results of calibration experiments using a series of polyethylene glycols (PEG) with known saturation pressures $P_{sat}$ (Pa) at reference temperature of 298.15 K (Bannan et al., 2018; Krieger et al., 2018). All reported $C^*$ and $P_{sat}$ values are thus contrasted to this temperature. The calibration method

and resulting parameters are described in the supplementary material Sect S3.





### 2.2.3 SOA yield calculations and theoretical yields

From the amount of consumed precursor VOC (*ΔVOC)*, and the condensed (i.e., particulate) organic mass (Δ$C_{OA}$) one can calculate the effective SOA mass yield *Y* by

$$Y = \frac{\Delta C_{OA}}{\Delta VOC_{MT+SQT}}. \tag{1}$$

The subscripts here refer to monoterpenes (MT) and sesquiterpenes (SQT), which were the dominant contribution to SOA formation in these experiments. The *ΔVOC*$_{MT+SQT}$ (μg m$^{-3}$) was calculated from the difference of the total concentrations of monoterpenes and sesquiterpenes entering the OFR and the VOC concentration exiting the OFR, which is quantified by the PTR-MS. Measured mixing ratios of both MTs and SQTs exiting the OFR were <1 ppb. The condensed organic mass Δ$C_{OA}$ (μg m$^{-3}$) was monitored using the SMPS number size distribution and assuming a particle density of 1.3 g cm$^{-3}$ for all
measurements (Faiola et al., 2018). No wall-loss correction was applied to the VOC and particle measurements. However, the same length of tubing was used for all the measurements so possible wall-losses are assumed to be nearly identical.

The SOA yield can also be expressed using the approach of Odum et al. (1996), which allows for formally breaking down SOA formation into the contribution of several VOC species:

$$Y = \Delta C_{OA} \sum_i \frac{\alpha_i K_i}{1+K_i \Delta C_{OA}}, \tag{2}$$

where $\alpha_i$ is a proportionality constant relating the amount of reacted VOC (precursor) to the total concentration of species *i* (oxidation product) and $K_i$ is the partitioning coefficient for species *i*. We use a two-product version of equation (2), which is common in modeling applications.

### 2.3 Scots pine sapling treatment

Due to scheduling constraints, the experiments were conducted in early autumn outside of the most active emission time for scots pines. To circumvent this, Scots pine saplings (6 years old) were stored in a cold room throughout the summer to maintain winter dormancy. They were removed from the cold room ~1 month before experiments to initiate spring phenology and metabolism – the time of year when Scots pines are active with new shoot growth. At least 24 hours before
the SOA experiments, saplings were transported to the laboratory and a dynamic Teflon plant enclosure was installed around the plant foliage. The Teflon enclosure (custom-built; Jensen Inert Products, Inc.) was secured to the plant stem with two cable ties. The plant enclosure was flushed with 5 lpm scrubbed/filtered compressed air and operated under positive pressure to push enclosure air into the flow reactor through PFA tubing. The flow of plant enclosure air into the OFR was regularly measured in-line and maintained at 1.7-1.9 lpm. Three LED lamps were placed around the plant to provide
photosynthetically active radiation (PAR).





## 3. Results and discussion

### 3.1 Scots pine emission patterns

The VOC emissions from the Scots pine sapling were mainly composed of monoterpenes and sesquiterpenes. The emissions
were monitored on-line with PTR-MS and off-line with TD-GC-MS, the latter allowing us to better speciate the VOCs by
distinguishing between isomers. In Fig. 2 we show the chemical structures of the most abundant compounds (see Table S2
for full list), along with the relative concentrations of the individual monoterpenes and sesquiterpenes, as measured by TD-
GC-MS.

Among the monoterpenes, β-phellandrene had the highest concentration, followed by 3-carene, d-limonene, β- and α-pinene,
in this order. These five monoterpenes accounted for 90 % of all monoterpenes in all experiments. Among the
sesquiterpenes, β- and α-farnesene, and α-bisabolene were the most abundant species, accounting for about 95 % of all
sesquiterpenes. All sesquiterpenes combined accounted for 55 % - 70 % of the total VOC mass concentration in the Scots
pine experiments 1-3, and for 40 % in the Scots pine experiment 4, based on PTR-MS measurements. Relating these results
to the ambient pine emission characterizations by Bäck et al. (2012), our Scots pine sapling may be classified as a "3-carene"
chemotype, due to the higher emission of 3-carene. The fraction of sesquiterpene emissions in our study was considerably
higher than expected from the ambient pine emission measurements of SQT/MT ratio of typically ~ 0.1 by mixing ratio
(Hellén et al., 2018). The used scots pine sapling was infested with herbivores creating biotic stress for the plant changing
the VOC emission pattern. This type of biotic stress is natural, especially in a changing climate where insect outbreaks are
predicted to become more frequent (Bale et al., 2002; Jactel et al., 2019).

During the Scots pine experiments 1-3, the VOC levels were slowly decreasing over time. Therefore, after the end of the
Scots pine experiment 3, we made four 0.5-1 cm$^2$ cuts to the Scots pine sapling's stem, with the goal of increasing VOC
emissions to continue the production of sufficient SOA. The wounds exposed plants resin pools in the stem and increased the
measured monoterpene concentrations. Sesquiterpene concentrations, however, were not significantly affected. The
compound that increased most was β-phellandrene, emissions of which had also been shown to increase with bark beetle
infestation damage (Amin et al., 2012; Faiola et al., 2018). Such damage essentially consists of cuts into the stem as the bark
beetle feeds on the plant, hence that observation is consistent with expectations.





### 3.2 SOA mass yields

In Figure 3, we plot SOA yields against condensed organic mass $C_{OA}$ from each experiment, split into two panels, according to which of the two PAM reactors was used (Fig. 3a, "PAM 1"; Fig. 3b, "PAM 2"). Note that a similar organic mass range was covered in all experiments.

The Scots pine experiments 1-3 featured sesquiterpene-to-monoterpene ratios (SQT/MT) by mass ratio between 1.2 and 2

(color-coding in Fig. 3a). The SOA yields obtained during these experiments are consistent with each other in a sense that all the data points are well described by the same two-product model (orange line). The SOA yields resulting from Scots pine experiment 4 are about 30% larger, even though SQT/MT was substantially smaller than in experiments 1-3.

To understand the unexpected suppression of SOA yield at higher relative SQT concentrations, we also measured SOA

yields for a synthetic sesquiterpene mixture containing isomers of farnesene and bisabolene. A different PAM reactor had to be used, therefore we compare those results to a reference α-pinene experiment using the same reactor (Fig. 3b). The ingoing VOC concentration was varied for both precursors and yield measurements were conducted in order to cover the $C_{OA}$ range of the Scots pine experiments. A single FIGAERO measurement was made for both precursors and the corresponding sampling conditions are marked with circles in Fig. 3b. The comparison shows a clearly lower yield for the sesquiterpene

mixture than for α-pinene, which is in line with results shown in Fig. 3a. Note that the medium and low exposure a-pinene measurements in PAM 1 show similar yield values as those in the reference experiment in PAM 2, but the yield of the high exposure experiment is significantly lower. This is consistent with the extensive fragmentation expected inside the PAM reactor under strong oxidative conditions which reduces the effective SOA yield (Lambe et al., 2012).

We conclude that the increase in SOA yield in the Scots pine experiment 4, compared to the Scots pine experiments 1-3, is likely due to the large relative increase in emitted monoterpenes, esp. β-phellandrene, caused by cutting the sapling (Fig. 2). We surmise that β-phellandrene must have a high SOA yield, possibly comparable to that of d-limonene, which has been reported in a range of 50-60 % (Berg et al., 2013; Lee et al., 2006; Surratt et al., 2008), and which shares some structural similarity with β-phellandrene (Fig. 2). Mackenzie-Rae et al. (2017) measured SOA yields of α-phellandrene, an isomer of β-

phellandrene with an endocyclic C=C bond and found the SOA yield to be around twice that of α-pinene, reaching up to 100 %. While those yield numbers are not directly comparable to our results, they qualitatively indicate that SOA yields from monocyclic monoterpenes, with endocyclic C=C bonds, could be higher than those from the more commonly studied bicyclic monoterpenes, such as α-pinene.

It is instructive to compare our SOA yield results to results of earlier experiments that used (stressed) Scots pines. Faiola et al. (2018) studied SOA formation from emissions of herbivore-stressed Scots pines in a custom-made OFR where the main



SQT was β-caryophyllene. There the increasing SOA yields with increasing SQT contribution to the precursor mix was explained by the much higher SOA yield of β-caryophyllene (Faiola et al., 2018). In chamber studies with emissions of aphid stressed Scots in which farnesenes were the dominant SQT species, SOA yields decreased with increasing SQT contribution

for ozonolysis reaction while no change was observed for pure OH reaction experiments (Faiola et al., 2019). In our experiments in the PAM reactor, the ratio between $O_3$ and OH exposure was in the range of $10^5$ which is comparable to ambient levels (Kang et al., 2007). Due to the very fast reaction with $O_3$, more than 80 % of any SQT is expected to react with $O_3$ under these conditions (calculated with methods described by Peng et al. (2015, 2016)). Thus, the SOA yield should decrease with increasing amounts of farnesene as described for the ozonolysis reaction pathway in Faiola et al. (2019).


The reason for this different behavior of the two SQT types is based on their molecular structure. β-caryophyllene is a bicyclic compound with one endo- and one exo-cyclic C=C bond whereas farnesene isomers are acyclic compounds with 4 C=C bonds. In the (photo-)oxidation process, both ozone and OH-radicals break the C=C bonds and thus the carbon back bone of farnesene into small fragments. Already a single oxidation step creates decreases the number of carbons from 15 to 5

- 12 (Kourtchev et al., 2009, 2012). In case of the bicyclic β-caryophyllene, such fragmentation is expected to be much less prominent (Jaoui et al., 2003, 2013).

There may be further interactions between the small, most likely volatile farnesene reaction products and the other oxidation products suppressing the particle formation further as recently described by (Mcfiggans et al., 2019).


### 3.3 SOA composition

Figure 4 shows mass spectra integrated over the heating period of FIGAERO-CIMS measurements normalized to the maximum signal and molecular mass adjusted to neutral compositions (Sect. 2.2.1). The results are grouped in two portions as two different PAM reactors and FIGAERO inlets were used in the measurements (Sect. 2.2). The α-pinene measurements

(Fig. 4, top row) show a decrease in the average molecular weight with increasing OH exposure, which is consistent with the decrease in SOA mass yields as a function of OH exposure observed in Fig. 3.

The distribution of molecular weights of compounds in the sesquiterpene mixture SOA particles (Fig. 4, right) shifted more towards smaller molecular masses than in any other experiment. We explain this by the acyclic molecular structure of the

sesquiterpene compounds leading to their more efficient splitting into smaller products (fragmentation) during oxidation (see Sect. 3.2). The α-pinene reference spectra differ slightly from other α-pinene measurements, even though OH-exposure falls between medium and high exposure (Table 2).



The difference might be due to the different residence time inside the PAM reactor (120 vs 160 s) or from the higher mixing
ratio of VOCs in the experiment done with the PAM2 reactor. The FIGAERO mass spectra distribution of SOA formed from
the Scots pine emissions had most similarities to high-exposure α-pinene SOA, with almost all compounds appearing within
a single mode, roughly centered on the monomer mode of the other α-pinene experiments.

Table 3 shows the average carbon oxidation states (OSc) (Kroll et al., 2011) and the average O:C ratio, calculated from both
AMS and FIGAERO data. The average chemical composition is also calculated from FIGAERO data. All FIGAERO data
values are weighted averages, using the integrated signal strength of each ion thermogram as weights. The range of values
calculated from the FIGAERO data represents the spread of different compounds in the mass spectrum, which we will
investigate below.

In the α-pinene experiments with PAM1, the average O:C and OSc increases with the increasing oxidative strength as
expected while the average carbon chain length decreases. In terms of the average O:C or OSc, the Scots pine experiments
most closely correspond to the medium exposure α-pinene experiment. Interestingly, all Scots pine experiments appear
broadly similar from this viewpoint, even though experiment 4 is associated with emissions dominated by monoterpenes
(i.e., much smaller SQT/MT) and clearly higher SOA yields (Figs. 2 and 3a). However, the Scots pine SOA particles differ
from each other and from other experiments in other ways, as we will see in Sect. 3.4.

### 3.2 SOA volatility

In Figure 5, we show normalized "sum" thermograms, calculated by summing up the thermograms from all ions that contain
only C, H and O atoms. With these sum thermograms, the average thermal desorption behaviour can be compared across
multiple experiments. While viewing the thermograms, it is important to know that the thermal decomposition during the
aerosol desorption from the FIGAERO filter often manifests as signal at relatively high desorption temperatures, which may
appear as shoulders to the main peak, but it can appear as simple peaks as well (e.g., Lopez-Hilfiker et al., 2015;
Schobesberger et al., 2018).

For the α-pinene experiments, the sum thermograms reveal a clear increase of $T_{max}$ (dashed vertical lines in Fig. 5a) with
increasing oxidative exposure. As we have shown above, this change is concurrent with a decrease in molecular size and an
increase in average O:C-ratio (Fig. 4 and Table 3). Together, these observations imply that α-pinene photo-oxidation
successively forms compounds with lower volatility. Evidently, the oxidation reactions are both fragmenting, which
generally increases volatility, as well as functionalizing, which generally decreases volatility (Capouet and Müller, 2006;
Pankow and Asher, 2008). The clearly increasing $T_{max}$ values observed with FIGAERO suggests a net decrease in volatility
due to these processes overall, for the condensed-phase constituents, consistent with the results of isothermal evaporation
experiments performed using the same (but size-selected) aerosol (Buchholz et al., 2019).





The sum thermograms for the Scots pine experiments gradually shift towards yet higher desorption temperatures (Fig. 5b); all their $T_{max}$ values are higher than those of any of the α-pinene experiments (Fig. 5a). In particular, SOA particles from

Scots pine experiment 4 are the most resistant to thermal desorption. That experiment was also the only one with plant emissions clearly dominated by monoterpenes (Fig. 2), specifically with β-phellandrene being the most abundant, which we associated above with relatively high observed SOA yields (Sect. 3.1). That specific VOC mix in the Scots pine experiment 4 is unique within our study here. It is plausible that this mix is also directly responsible for producing SOA with the lowest effective $C^*$ as suggested by the results of our FIGAERO measurements (Fig. 5, as well as our more detailed discussion

below).

We suggest that the increased desorption temperatures for the Scots pine experiments 1-3 relative to the α-pinene experiments is due to the large contribution of acyclic sesquiterpenes (in particular farnesenes, Fig. 2) to the plant emissions in those experiments. We tested this hypothesis via our follow-up experiments using the PAM 2 reactor and FIGAERO 2

(Fig. 5c). These experiments yielded sum thermograms for SOA particles formed from the farnesene-dominated sesquiterpene mixture, and those from α-pinene using the same setup, for reference. The reference α-pinene SOA particles had a $<O:C>_{AMS}$ of 0.77, similar to those in the medium and high exposure α-pinene case. Indeed, the sum thermogram $T_{max}$ value for the sesquiterpene mix case is about 10 °C higher than for the reference α-pinene case, confirming that the strong contribution of farnesene and sesquiterpenes with similar structures leads to the effectively lower-volatility aerosol particles

(as measured by FIGAERO) in the Scots pine experiments 1-3 compared to α-pinene SOA particles. As mentioned earlier (Sect. 2.2), a quantitative comparison of thermograms between Fig. 5c and Figs. 5a-b, including comparison of $T_{max}$ values, is not straightforward, due to the differences in the respective experimental setups. However, we will deal with this issue below.

We provide a more extended discussion of our examination of SOA volatilities, which includes graphical depictions of the individual ion thermograms, in the supplement material (Sect. S4). When looking at individual experiments, the $T_{max}$ value for each ion broadly depends on the molecular weight of the ion, as expected. However this dependency seems to differ in scale between experiments. Some signatures of thermal decomposition are visible as well, but overall this appears to play a minor role, with very small effects on $T_{max}$ in most individual ion cases. Consequently, the shifts to higher desorption

temperatures as observed in the sum thermograms (Fig. 5) are essentially seen throughout each respective spectrum of individual unit mass thermograms, although the contribution of thermal decomposition appears to increase concurrently.

Previous studies have indicated that the $T_{max}$ value of individual thermograms largely remains controlled by the $C^*$ of the respective compound, even when a substantial fraction of the signal is the result of thermal decomposition of different, larger

structures (e.g. Schobesberger et al., 2018). The reason is that this decomposition typically occurs only at sufficiently higher





temperatures than the desorption temperatures for most compounds. Measured $T_{max}$ values can therefore be used as a fairly robust estimation for $C^*$ of the respective composition. Thus, we performed calibration experiments, in order to establish the $T_{max}$-$C^*$ relationship for both FIGAERO inlets used in this study (see Sect. S3), and accordingly derived $C^*$ for each measured organic composition from its respective $T_{max}$ value. Note that in cases where thermogram peaks are affected by

thermal decomposition (which we determined to play a relatively minor role in this study), we thereby implicitly assign upper-limit $C^*$ values. We also note that relatively high collected aerosol mass (in the order of 1 μg) might induce so-called matrix effects in the evaporation process (Huang et al., 2018), which might in turn shift the observed $T_{max}$ to higher temperatures, and could consequently lead to a slight systematic underestimation of $C^*$.

The results from the $T_{max}$-$C^*$ conversion are shown in Fig. 6 and 7, summarizing them as Volatility Basis Sets (VBS) bins of one order of magnitude of $C^*$, as defined by Donahue et al., 2011 (SVOC = Semi-volatile Organic Compounds, green volatility range; LVOC = Low Volatility Organic Compounds, red volatility range; ELVOC = Extremely Low Volatility Organic Compounds, grey volatility range). These VBS distributions represent only the particle phase. Note that having translated from $T_{max}$ to $C^*$ using the instrument-specific calibration parameters, all results become comparable to each other.

Also note that the abundance of SVOC compounds might generally be underestimated due to the fast evaporation of particulate matter during collection to FIGAERO filter and switching from collection phase to desorption phase.

A clear shift from higher to lower volatility for α-pinene SOA is observed with increasing OH exposure (Fig. 6). This shift manifests in the disappearance of SVOC compounds while especially the amount of LVOC compounds increases. For

sesquiterpene mixture SOA particles, a major part of the observed compounds falls into the LVOC class, while SVOC class compounds are almost non-existing. This is also seen in the α-pinene reference experiment, where most of the compounds are attributed to a single VBS bin. Note also the different y-scale in the α-pinene reference results compared to other results.

In the Scots pine results (Fig. 7), a similar shift from higher to lower volatilities can be seen, with the Scots pine experiment

4 having the lowest volatility overall. Differences in the Scots pine VBS are probably due to evolving VOC emissions from the sapling. Note that simple SQT/MT ratio calculated from total amount of monoterpenes and sesquiterpenes (Table 1, insets in Fig. 8 top row) cannot explain the change in the volatility.

Finally, we examine the compositional information present in the FIGAERO datasets. In Fig. 8, we plot the carbon numbers

versus oxygen numbers for each compound for each experiment, coloured according to the $T_{max}$-derived saturation concentration ($C^*$). The panels reveal the spread of compounds across the ranges of O:C ratios. In the α-pinene and sesquiterpene mixture experiments, the compounds fall roughly into similar areas (i.e., range of O:C ratios: 0.5 – 1.0), but in the Scots pine experiments the spread across different O:C ratios is notably wider. Specifically, there is an increased contribution of both relatively small (#C ~ 5) but highly oxygenated compounds and larger (C>10) though less oxygenated





compounds (O:C < 0.5), both with relatively low effective volatility. This suggests that the bulk O:C ratio is insufficient to compare the expected properties of SOA particles generated from mixtures of different VOC precursors (e.g. MT and SQT, cf. average O:C and OSc values with standard deviations in Table 3.) Partially, the profoundly different spread of O:C ratios in the Scots pine experiment results might also be due to more extensive thermal fragmentation in the FIGAERO desorption process.


## 4. Summary and conclusions

In this study, we compared the physicochemical properties of SOA particles generated from combined ozonolysis and photo-oxidation of (1) α-pinene, (2) a complex mixture of VOCs emitted from Scots pine saplings, and (3) a mixture of farnesenes and bisabolenes which were identified to contribute a significant fraction to the Scots pine emissions. Our measurements

examined the SOA mass yield, as well as the chemical composition and the thermal desorption properties of SOA particulate constituents to assess their volatility. In general, we found that all of these quantities or properties are substantially controlled by the amount of OH exposure, as expected, but crucially also by the identity of the isomers making up the precursor VOC mixture.

The SOA mass yield from Scots pine emissions was in general lower than the yield from α-pinene. The notable exception is a single experiment (Scots pine experiment 4) in which plant emissions were not dominated by farnesene and sesquiterpenes with similar structure, but by β-phellandrene and other monoterpenes. That experiment had the highest SOA yields among the Scots pine experiments, which is consistent with the high yields for the specifically involved monoterpenes reported or suggested in the literature (e.g. Faiola et al., 2018). For the other Scots pine experiments (experiments 1-3), we attribute the

low observed SOA yields to the substantial contribution of acyclic sesquiterpenes, particularly β- and α-farnesene, to the total terpenoid plant emissions in those cases, ranging from 40-70% by mass. This is supported by our additional experiments that resulted in a much lower SOA yield from a mixture of mono and acyclic sesquiterpenes than from α-pinene. When such acyclic compounds are oxidized, the initial reaction with OH or $O_3$ already leads to fragmentation (i.e. reaction products with much shorter C chain), whereas cyclic compounds, such as many monoterpenes, (including α-pinene

and β-phellandrene) but also bicyclic sesquiterpenes (such as β-caryophyllene), generally go through more steps of oxidation before the onset of significant fragmentation is observed. Such enhancement of the fragmentation processes of acyclic sesquiterpenes likely result in a product distribution containing a smaller amount of organic materials with sufficient low volatility to partition from the gas to particle phase. These acyclic sesquiterpenes might also go through multiple cycles of auto-oxidation following the reaction with OH, before suffering substantial fragmentation (Bianchi et al., 2019), which

would explain the relatively high (>5) amount of oxygen atoms in a major fraction of the SOA compositions.





Indeed, our thermal desorption results indicate that the oxidation of acyclic and monocyclic sesquiterpenes forms a substantial amount of relatively small compounds, compared to α-pinene (Fig. 4). The average molecular weight of particle-phase α-pinene oxidation products decreases with increasing OH exposure (Table 3), while their O:C ratio increases. At the

same time, the thermal desorption temperature of the SOA particulate constituents increased (Figs. 5-6), indicating a decrease in the effective SOA particle volatility.

Interestingly, our measurements showed that our Scots pine and sesquiterpene SOA particles were of lower volatility than any of the α-pinene SOA particles (even at higher oxidation exposure and comparable O:C/OS$_C$ values). This result is

indicated by both sum thermograms and supported by their derived VBS distributions for the individual SOA particulate constituents. It is also worth noting that, for the Scots pine experiments, the lowest-volatility SOA is formed in the experiment resulting in the highest SOA yields (Scots pine experiment 4), contrary to the observations made for the α-pinene experiments.

The molecular composition and thermal desorption behaviour of SOA particles observed with the FIGAERO instruments in that experiment 4 were strikingly similar to the other, sesquiterpene-dominated, Scots pine experiments 1-3. It appears that a certain contribution of those acyclic sesquiterpenes is sufficient to low SOA volatility, whereas the monoterpenes that dominate the mixture led to the efficient formation of SOA to start with. Interestingly though, the monoterpenes did not seem to directly affect the volatility of Scots pine SOA, at least for our experiments here, even though their relative

contribution to the precursor VOC mixture varied substantially. Further experiments are clearly warranted to explore this suggestion for a wider range of conditions and precursor mixtures than covered by this study.

In conclusion, our results highlight the need of knowing the structural identity of mixtures of VOCs, as typically encountered in real atmospheric conditions, if one endeavours an accurate prediction of SOA yields and SOA particle properties. In

particular the emissions of sesquiterpenes need to be considered more carefully in current atmospheric models. Importantly, their effects on SOA yields can be both enhancing and suppressing, depending on the involved isomers. Specifically, depending on which sesquiterpene isomers are involved, the product distributions obtained from their oxidation can differ substantially from each other in terms of the products volatility and of the subsequent SOA chemistry. At the very least, a differentiation between cyclic and acyclic terpenes is desirable. This issue is likely to become more relevant in the future,

when biological and abiotic plant stressed events increase in frequency as is projected for a warming climate (Bale et al., 2002; Jactel et al., 2019). Such stresses will both increase biogenic VOC emissions and change the composition of emitted mixtures (Faiola et al., 2018, 2019).

*Data availability.* The data shown in the paper is available on request from corresponding author.




*Author contributions.* AV, CF, AB and TY-J designed the study. AY, AB, ZL, LB, AL, CF, EK and SN performed the measurements. AY and SS led the paper writing and all of the co-authors participated to the interpretation of the results and paper editing.

*Competing interests.* The authors declare that they have no conflict of interest.

*Acknowledgements.* The authors wish to thank James Blande, Minna Kivimäenpää and Rajendra Ghimire (University of Eastern Finland, Department of Environmental and Biological Science) for tending the Scots pine seedlings. Sergey A. Nizkorodov acknowledges the Fulbright Finland Foundation and the Saastamoinen Foundation that funded his visit to the
University of Eastern Finland.

*Financial support.* This research was supported by Academy of Finland (272041, 310682, 299544), European Research Council (ERC-StQ QAPPA 335478) and University of Eastern Finland Doctoral Program in Environmental Physics, Health and Biology.

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




**Table 1. List of different Scots pine experiments with corresponding PAM 1 reactor conditions. Monoterpenes are referred to as MT and sesquiterpenes as SQT. Collected mass is estimated from SMPS data assuming particle density of 1.3 g cm$^{-3}$.**

| Experiment | Scots pine 1 | Scots pine 2 | Scots pine 3 | Scots pine 4 |
|---|---|---|---|---|
| VOC mixing ratio (ppb from PTR-ToF-MS) | MT: $125 \pm 1$<br>SQT: $179 \pm 1$<br>Sum: $304 \pm 2$ | MT: $213 \pm 14$<br>SQT: $179 \pm 1$<br>Sum: $391 \pm 14$ | MT: $127 \pm 4$<br>SQT: $151 \pm 1$<br>Sum: $278 \pm 4$ | MT: $153 \pm 3$<br>SQT: $66 \pm 2$<br>Sum: $218 \pm 3$ |
| SQT/MT-ratio (by molar ratio) | 1.43 | 0.85 | 1.2 | 0.4 |
| SQT/MT-ratio (by mass ratio) | 2.15 | 1.26 | 1.78 | 0.65 |
| OH-exposure (molecules cm$^{-3}$s) | $6.41 \times 10^{11}$ | $6.49 \times 10^{11}$ | $6.45 \times 10^{11}$ | $7.31 \times 10^{11}$ |
| PAM residence time (s) | 300 | 300 | 300 | 300 |
| O$_3$ mixing ratio (ppm) | 4.9 | 5 | 5 | 5.6 |
| Collected mass on FIGAERO filter (ng) | 1000 | 1200 | 1300 | 1000 |






**Table 2. List of different α-pinene and sesquiterpene experiments with corresponding PAM-reactor conditions. Collected mass is estimated from SMPS data assuming particle density of 1.3 g cm$^{-3}$**

| Experiment | PAM 1 | | | PAM 2 | |
|---|---|---|---|---|---|
| | α-pinene low exposure | α-pinene medium exposure | α-pinene high exposure | Sesquiterpene mixture | α-pinene reference |
| VOC mixing ratio (ppb from PTR-ToF-MS) | $199 \pm 2$ | $198 \pm 2$ | $196 \pm 2$ | $327 \pm 5$ | $270 \pm 10$ |
| OH-exposure (molecules cm$^{-3}$ s) | $2.54 \times 10^{11}$ | $6.85 \times 10^{11}$ | $2.45 \times 10^{12}$ | $8.2 \times 10^{10}$ | $2.6 \times 10^{11}$ |
| PAM residence time (s) | 120 | 120 | 120 | 160 | 160 |
| O$_3$ mixing ratio (ppm) | 6.6 | 25 | 25 | 13.3 | 13.2 |
| Collected mass (ng) | 960 | 880 | 1350 | 550 | 925 |






**Table 3. Average O:C ratio, OSc and chemical composition from each experiment calculated from AMS and FIGAERO data. Values calculated from FIGAERO are weighted by the integrated signal strength. All FIGAERO data is shown with standard deviation to highlight the spread of different compositions in the mass spectrum.**

| Experiment | $<O:C>$ AMS | $<O:C>$ FIGAERO | $<OSc>$ AMS | $<OSc>$ FIGAERO | $<C_xH_yO_z>$ |
|---|---|---|---|---|---|
| α-pinene low exposure | 0.53 | 0.65±0.28 | -0.46 | -0.3±0.56 | $C_{9±4} H_{14.2±4} O_{5.4±4}$ |
| α-pinene medium exposure | 0.69 | 0.75±0.3 | -0.05 | 0.02±0.65 | $C_{8.3±3.5} H_{12.1±3.5} O_{5.7±3.5}$ |
| α-pinene high exposure | 0.96 | 0.9±0.33 | 0.63 | 0.47±0.73 | $C_{7.5±3} H_{9.9±4.5} O_{6±1.9}$ |
| Scots pine experiment 1 | 0.85 | 0.79±0.43 | 0.37 | 0.02±1.1 | $C_{8.5±4} H_{13.6±8} O_{5.4±2}$ |
| Scots pine experiment 2 | 0.89 | 0.77±0.42 | 0.43 | -0.04±1.1 | $C_{8.4±4} H_{13.5±8.3} O_{5.4±2.1}$ |
| Scots pine experiment 3 | 0.9 | 0.79±0.43 | 0.55 | 0.02±1.1 | $C_{8.7±4} H_{14±7.9} O_{5.4±2.1}$ |
| Scots pine experiment 4 | 1 | 0.81±0.43 | 0.75 | 0.06±1.1 | $C_{8.2±3.7} H_{13.1±7.5} O_{5.3±2}$ |
| Sesquiterpene mixture | 0.82 | 0.89±0.37 | 0.26 | 0.47±0.83 | $C_{7.5±3.7} H_{9.7±5.1} O_{5.6±1.8}$ |
| α-pinene reference | 0.77 | 0.82±0.33 | 0.06 | 0.28±0.71 | $C_{7.4±3} H_{10±4.4} O_{5.3±1.5}$ |




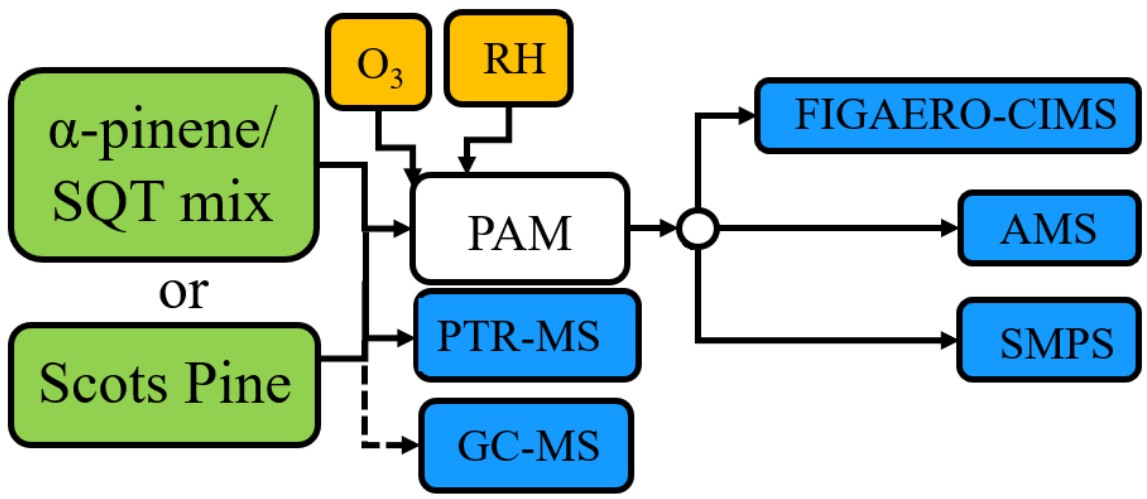

**Figure 1: Measurement setup used in the experiments. Abbreviations in the picture are explained in the main text (Sect. 2), except for ozone-containing air (O3) and humidified air (RH).**




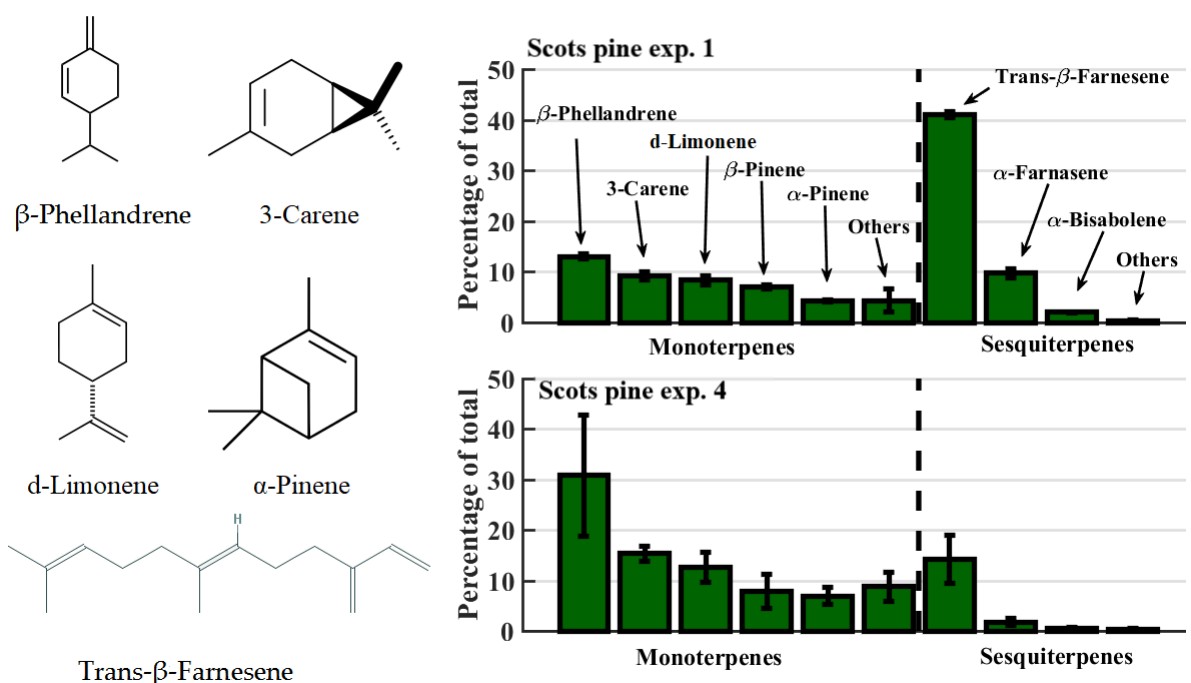

**Figure 2: Left panel shows the structures of the most abundant monoterpenes and sesquiterpenes in Scots pine emissions, as measured with TD-GC-MS. Relative mass concentrations of each compound are shown in the right panel for Scots pine experiments 1 (top) and 4 (bottom), the latter following deliberate damage to the plant's stem. The whiskers show the standard deviation of the measurements.**





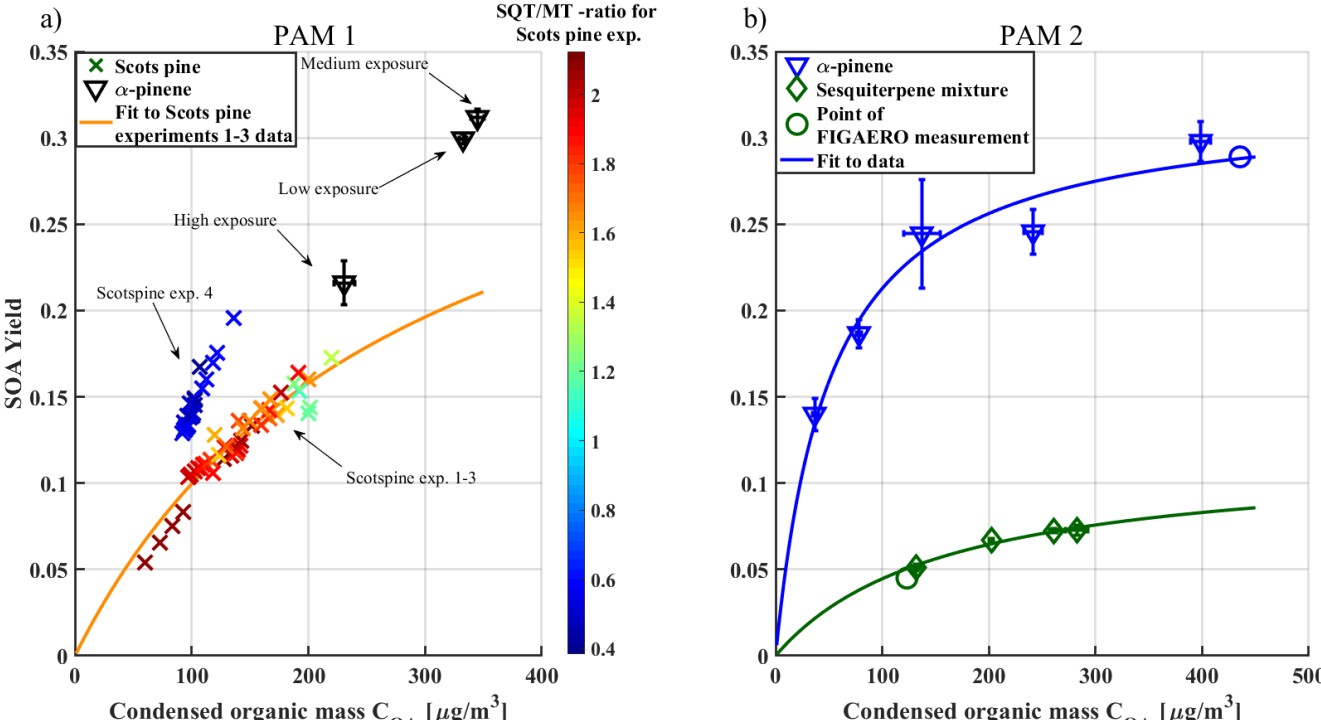

**Figure 3:** Panel a) SOA yield vs. Condensed organic mass ($C_{oa}$) from Scots pine and α-pinene experiments. Each point represents a single SMPS scan. In these cases, and error bars are omitted for clarity. The color scale on the Scots pine experiment results corresponds to the sesquiterpene-to-monoterpene (SQT/MT) ratio by mass ratio (crosses; showing all yield measurements, reflecting the variability in plant emission rates). Each α-pinene experiment is averaged to one point (black triangles) and labelled. Error bars shown are standard deviations of multiple scans. The orange curve is fit with equation (2) to the Scots pine experiments 1-3 (SQT/MT > 1) with fit parameters: $α_1 = 0.3783$, $K_1 = 0.0035$, $α_2 = 0.0029$ and $K_2 = 0.0035$. Panel b) α-pinene reference and sesquiterpene mixture SOA yields versus $C_{OA}$ and Odum fits to both datasets. The fit parameters are for α-pinene: $α_1 = 0.3219$, $K_1 = 0.0195$ and for the sesquiterpene mixture: $α_1 = 0.1164$, $K_1 = 0.0062$. Circles shows the points of FIGAERO measurement. Titles "PAM 1" (panel a) and "PAM 2" (panel b) refer to different PAM reactors used in the experiments.





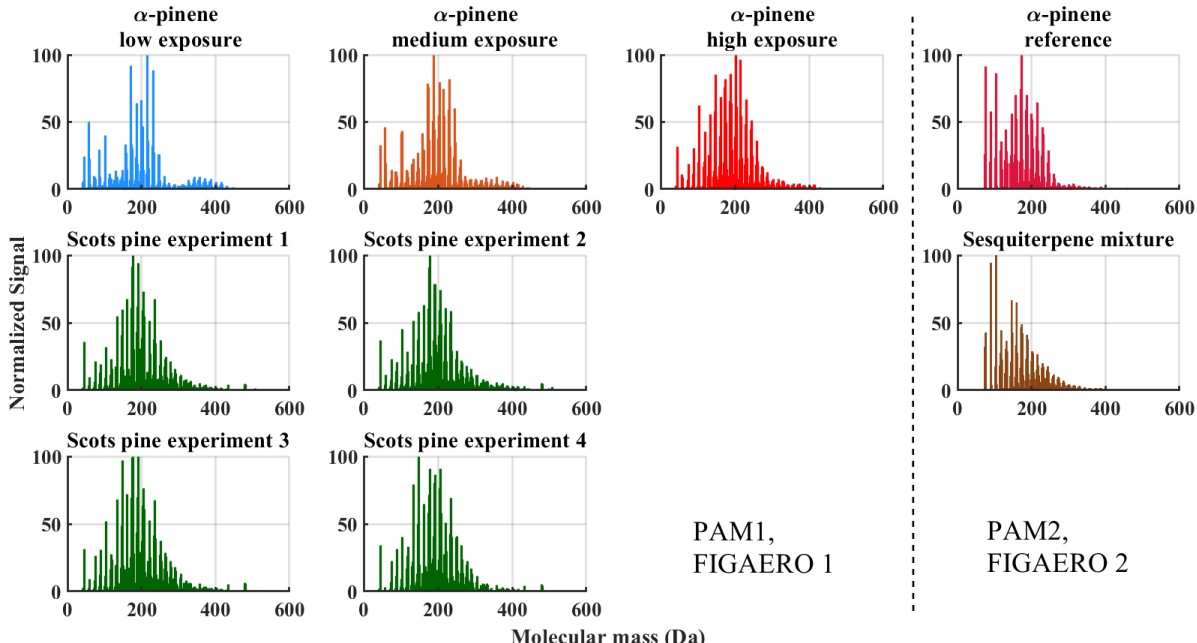

**Figure 4: Mass spectra integrated over the whole FIGAERO desorption cycle for each experiment. Each spectrum is normalized to the height of the most abundant ion.**




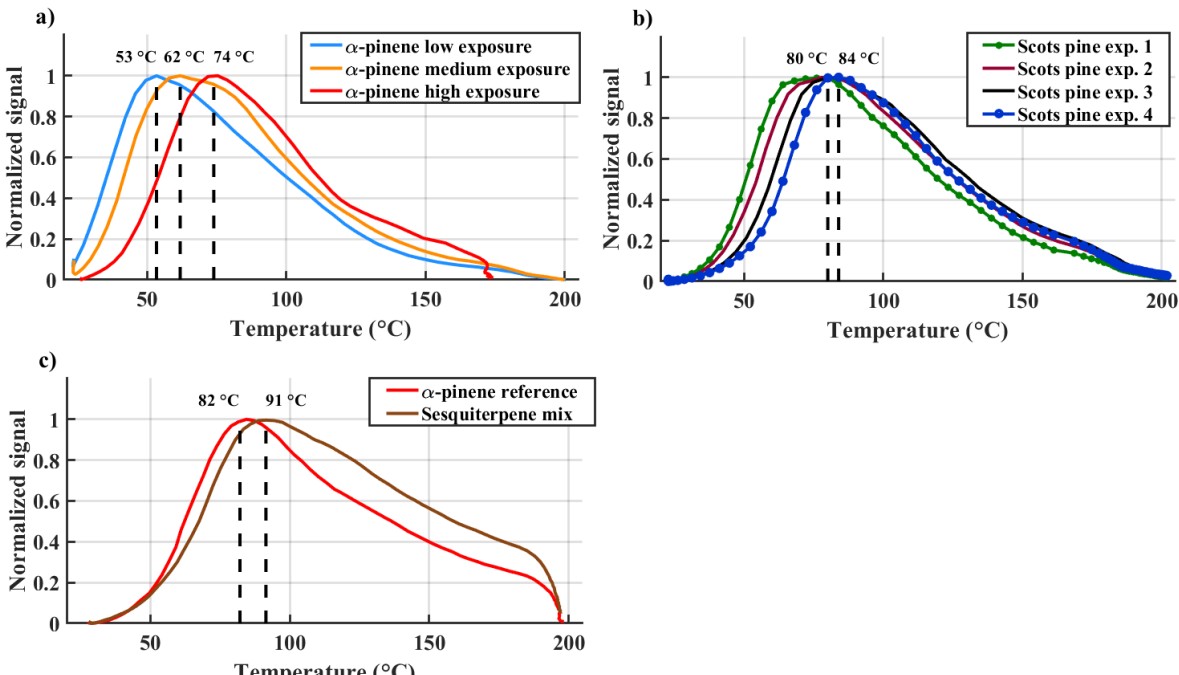

**Figure 5: Normalized sum thermograms (all observed ions containing C, H, and O atoms) from a) α-pinene experiments, b) Scots pine experiments 1 - 4 and c) sesquiterpene mixture experiment, together with reference α-pinene experiment.**




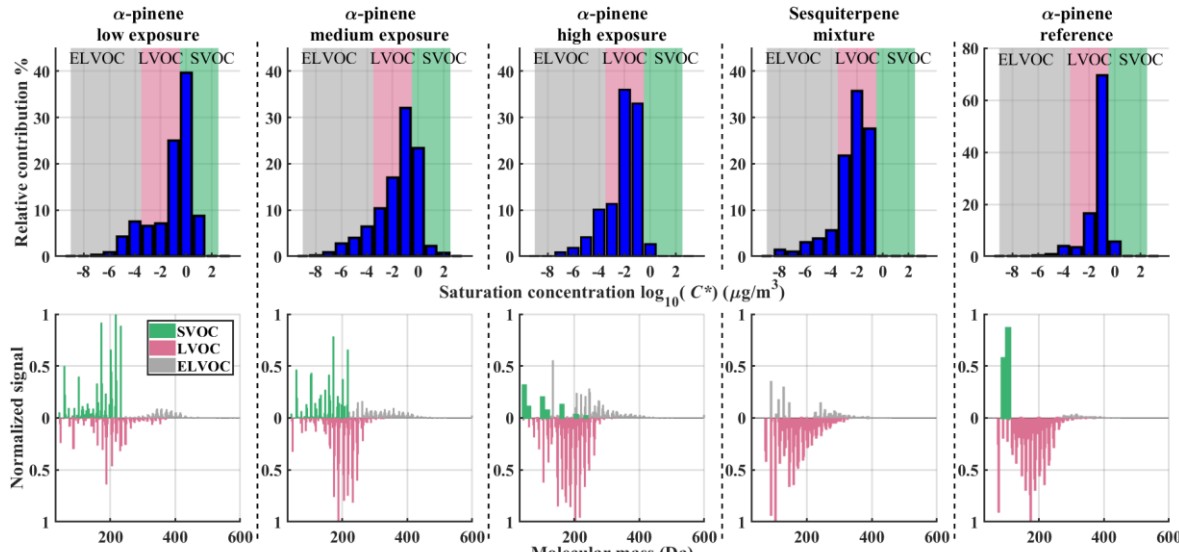

**Figure 6: VBS bins and corresponding normalized integrated mass spectra from α-pinene low, medium and high experiment, sesquiterpene mixture, and α-pinene reference experiment. Top row: VBS bins determined from *Tmax* with background colours corresponding to different volatility classes (defined in the text). Bottom row: Integrated FIGAERO signal normalized to maximum signal and coloured by corresponding volatility class. LVOC class compounds (red) are plotted on an inverted y-axis for easier distinction from other volatility classes VBS.**




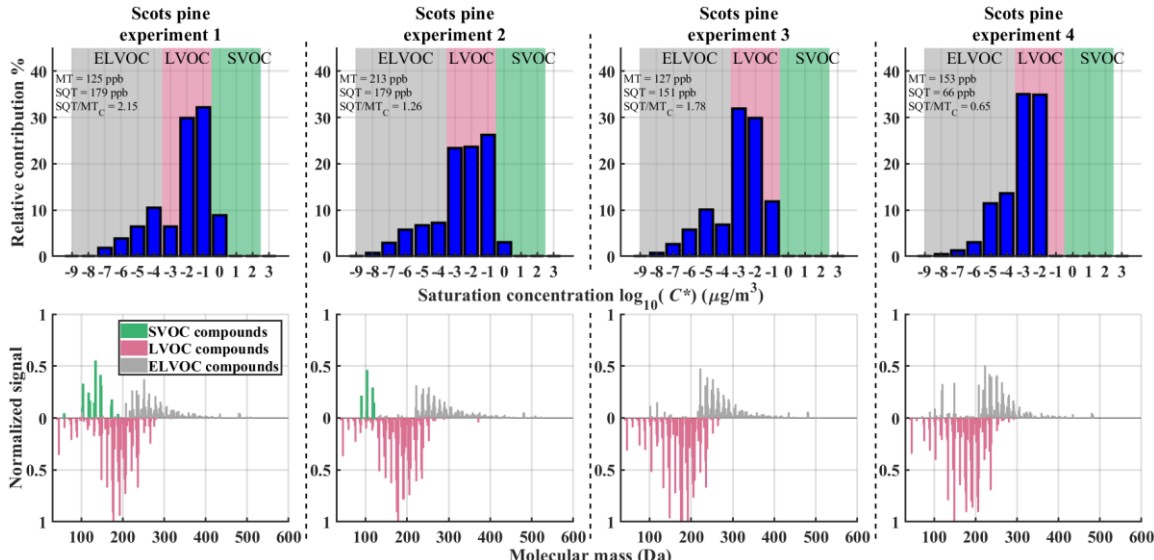

**Figure 7: VBS bins and corresponding normalized integrated mass spectra in the Scots pine experiments. Top row: VBS bins**
**determined from $T_{max}$ with background colours corresponding to different volatility classes. Bottom row: Integrated FIGAERO**
**signal normalized to maximum signal coloured by corresponding volatility class. LVOC class is plotted on reverse y-axis for easier**
**distinction from other volatility classes. Text in the upper row panels show measured amounts of monoterpenes and sesquiterpenes**
**and calculated SQT/MT ratio per mass ratio.**






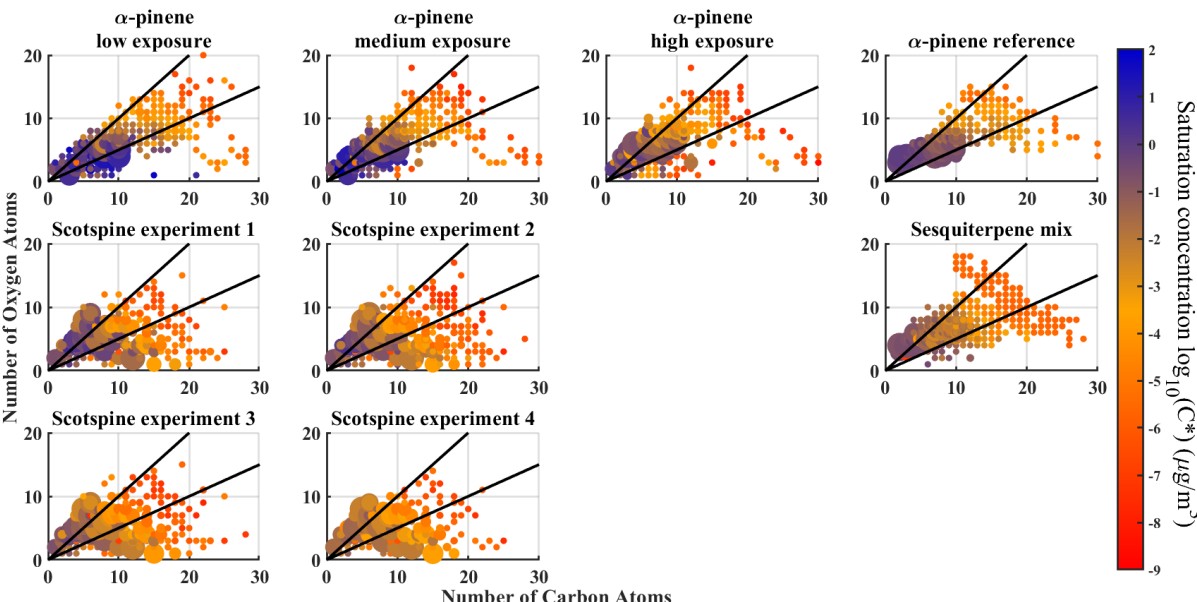

**Figure 8: Number of oxygen molecules versus number of carbon molecules for each compound and each experiment. Colours correspond to the saturation ratio $C*$ as derived from the measured $T_{max}$. The size of a marker corresponds to the signal of the compound. The two black lines in the figures correspond to O:C = 0.5 and O:C = 1 ratios, for reference.**
