# Peer review of "Composition and volatility of SOA formed from oxidation of real tree emissions compared to simplified VOC-systems"

_Atmospheric Chemistry and Physics, 2019_

## Referee Comment (RC1) · Anonymous Referee #1 · 22 Nov 2019

Review Ylisirniö et al., ACPD 2019, "Composition and volatility of SOA formed from oxidation of real tree emissions compared to single VOC-systems"

Ylisrniö et al. present a laboratory chamber study on the gas-phase oxidation of biogenic volatile organic compounds from both, pure VOC standards and real plant emissions. The work focusses on the effects of different monoterpene and sesquiterpene precursors (and their ratios) on SOA yield and volatility. They use adequate instrumentation to investigate (1) the VOC precursors by PTR-MS and GC-MS, (2) the secondary organic aerosol mass by SMPS and AMS, and (3) the volatility and molecular chemical composition by FIGAERO-CIMS. Although, there is an intrinsic problem with the

[Figure]

FIGAERO by using a thermal evaporation technique for the detection of highly labile organic compounds (HOMs, hydroperoxides, hydroxy-hydroperoxides, organic peroxides), the technique produces thermograms, which the authors interpreted with the necessary caution.

Studying SOA yields and how physicochemical properties of SOA changes with different single precursors, precursor mixtures, and real plant emissions, enables evaluating chamber experiments against the natural complexity. This is an important work for atmospheric chemistry and hence the paper is clearly within the scope of ACP. As mentioned above, the applied methods are appropriate and the introduction refers to other related work in the field.

However, as outlined below (major comments), the methods section leaves the reader with multiple questions on how the experiments were carried out in detail. Hence, the experimental description is not sufficiently complete and precise allowing for reproduction by other scientists/groups. I think that this can be fixed by going more into detail in the experimental part.

The overall quality of the graphs is good and the language is fluent and precise. The conclusions section should be more on the point in order to leave a clear message. Overall, I can recommend the paper to be published in ACP after addressing the following aspects and improving the information content in the methods section.

Major comments:

l. 25: Bisabolenes also have endocyclic double bonds. Their oxidation would not lead to smaller and more volatile compounds. I suggest rephrasing to "[. . .] is due to exocyclic C=C bond scission [. . .]"

l. 62: not only mixtures of VOCs influence the composition of products, but also trace gases like NOx and CO. Furthermore, ammonia and H2SO4 can alter particle phase composition through heterogeneous reactions with organic molecules. It would be

good to cite a review at this point (e.g. Ng et al., ACP, 2017, doi:10.5194/acp-17-2103-2017 for the NOx effects).

l. 82: Was the temperature actually monitored/regulated during the experiments, or does 25 °C just mean that experiments were done in the laboratory? I can imagine that the temperature changes in the flow tube when UV lamps are switched or when UV exposure is modulated.

Section 2.1: A more thorough description of the OFR-experiments is urgently needed. With the current description, it would be impossible to replicate these experiments. Instead of mentioning already results in the methods section (l. 84/85, l. 101), the level of information on the setup in the current version is unsatisfactory. As a reader I like to know for example:

How were blank experiments conducted? Was there a carry-over between different experiments when VOC precursors were changed? Did VOCs / SVOCs come back from the wall when switching on UV lamps? What were the flow rates? How were the VOCs flushed into the chamber (material of the tubing?)? Were the source- and sampling-lines heated to avoid condensation of sesquiterpenes? . . .

l. 107: What does the different "use history" of the PAM chambers mean? That statement leaves the reader questioning whether the history of experiments has an effect on the results?

Section 2.2: The level of experimental detail is unsatisfactory. A "suite of instruments" is mentioned in l. 112-115, but the following paragraphs only describe FIGAERO-CIMS and nothing on AMS or SMPS measurements. Concerning AMS, more information is needed in this experimental part, e.g. on the version (HR-AMS or C-ToF), operation mode (V-mode or W-mode) of the AMS, with/without size dependent composition, etc.

Furthermore, key information of FIGAERO-CIMS operation mode are also missing: how were blank measurements conducted? Have the authors evaluated whether gasphase adsorption on the particle filter can be an issue? What was the sample flow rate and the duration of particle collection? Until which maximum desorption temperature the ramp was operated? What was the pressure in the ion molecular region? etc.

Figaero data analysis: Why did the authors use tofTools and not Tofware, which offers also a figaero-version? Concerning peak identification: What are the numbers of allowed elements for C, H and O? How were multiple possible sum formulas for one exact mass ranked? Were ratios of H/C or O/C and double bound equivalents used to further constrain sum formula?

l. 167: Was an experiment conducted to measure wall/sampling line loss rates of sesquiterpenes against monoterpenes? E.g. SQT/MT measurements before and after the OFR, without O3 and UV off? L. 171: I disagree with this statement: wall/sampling line losses will be larger for sesquiterpenes than for monoterpenes. Relating this to the observed results, evaluating potential different losses of SQT compared to MT are highly relevant for the conclusions on different SOA yields.

Based on Fig. 2: does it need to be reassessed whether a-pinene is the (globally) strongest emitted monoterpene?

l. 274-276: this has already been shown by Hall et al. (https://doi.org/10.1021/es303891q)

l. 309-316: Is this observation in line with what one would expect from C* estimates based on the sum formula? (Donahue et al., ACP, 2011). It is surprising that oxygenation outweighs fragmentation of e.g. C20-dimers. Does the VBS distribution look different when the VBS bins are filled based on T_max vs. when they are filled based on the sum-formula-derived volatility?

Fig. 6 – SQT mix – mass spec: What are the ELVOCs with molecular mass < 200 Da? Are such compounds formed by thermal fragmentation of larger molecules? If this is the case, a quantitative estimate of the fraction of ion signal that results from thermal

fragmentation would help to evaluate the FIGAERO mass spectra.

l. 418: A comparison FIGAERO spectrum between acyclic and mono/bicyclic SQT would be very interesting. Fig. 4 only shows the spectrum of a SQT mix. More fragmentation products would be expected for the acyclic SQTs.

Minor:

l. 54 add Yassaa et al., ACP, 2012; doi:10.5194/acp-12-7215-2012

l. 119: oxygen or OH-groups: consider to be more specific. Would an ether (a molecule with an oxygen) already give a stable cluster with iodide?

l. 409: Explaining what fragmentation means should not appear in the summary / conclusions.

Technical:

l. 52 replace a-pinene with $\alpha$-pinene.

l. 62, l. 269 replace Mcfiggans with McFiggans.

Fig. 2: structural formula of farnesene explicitly shows one hydrogen. Either draw all hydrogens or none (preferably none).
* * *

---

## Referee Comment (RC2) · Anonymous Referee #2 · 24 Nov 2019

General Comments

The laboratory study of Ylisirniö et al. investigates the oxidation of monoterpene and sesquiterpene emissions from real Scots pine plants and contrasts them to synthetic mixtures. Gaseous emissions are oxidised in an oxidative flow reactor and subsequent SOA yields, chemical compositions and volatilities are compared. Regarding these properties, the importance of chemical structure for sesquiterpenes is highlighted specifically, as this can affect whether the precursor is SOA enhancing or supressing. The findings of this study highlight the complexities of studying real VOC mixtures compared with synthetic surrogates and in doing so makes an excellent contribution to

further current understanding in atmospheric chemistry.

The methods used are of a good quality, sufficient to describe the systems being investigated and support the conclusions that are drawn. However the methods section requires more detail in terms of instrument and experimental descriptions to ensure replicability.

The presentation of the contents is well thought out and shows a clear process of thought. Although effort has been made to keep extraneous detail to a minimum in order to make the manuscript concise, some parts may benefit from the addition of more information to improve clarity and emphasize the key findings. In some instances, the reduced clarity makes it difficult to understand the statement being made. In such cases it may be beneficial to reword the text.

This manuscript should be considered for publication in ACP after addressing the following comments:

Major Comments

Line 26. "SOA particles from the oxidation of Scots pine emission had similar or lower volatility than SOA particles formed from either of single precursor." Does the single precursors refer to the SQT mixture as well as the alpha pinene? If so that might need rephrasing. This is also true for the title.

Line 30. You state "These results emphasize that simple increase or decrease of relative monoterpene and sesquiterpene emissions should not be used as indicator of SOA particle volatility" but on line 27 you state "Applying physical stress to the Scots pine plants increased monoterpene emissions, which further decreased SOA particle volatility and increased SOA mass yield." Can you clarify or rephrase the statement on line 30 based on the statement from line 27.

Line 87. Could you please provide a brief description on the dynamic dilution system you use to mix your VOCs.

Section 2.2.1/S3 There is some missing information that would be useful. Can you please state the resolution of the ToF-CIMS. A description of your FIGAERO backgrounding methodology and how you account for backgrounds in your analysis is missing. Also a description of your gas and particle sample lines to the ToF-CIMS (material, length, residence time etc).

Line 169. Although you did not correct for wall losses, can you make an estimation or assessment of their importance? This may be important if the yields from this paper are compared with other experiments.

Line 186. Can you provide some more detail on the scrubber / filter for the compressed air.

Line 190. Can you provide more information on the LED lamps.

Line 210. Is the stress response induced by cutting the sapling reduced or altered because it is already stressed by the infestation? Are there any further considerations required as to how these two stresses alter VOC emission regarding the conclusions you can draw?

Line 214. Is it shown that MT concentrations were higher? Experiment 2 shows the highest [MT] concentration (table 1). Do you mean the transition from experiment 3 to experiment 4 specifically?

Line 227. " . . . are about 30% larger, even though SQT/MT was substantially smaller than in experiments 1-3." I find the words 'even though' confusing, can you explain why these findings are unexpected?

Line 241. "We conclude that the increase in SOA yield in the Scots pine experiment 4, compared to the Scots pine experiments 1-3, is likely due to the large relative increase in emitted monoterpenes" I think more clarification is needed regarding this conclusion. Can you provide a concise explanation of which factors regarding the MT are important for the increased SOA yield in experiment 4, such as absolute [MT] increase induced

by cutting; MT composition change induced by cutting e.g. $\beta$-phellandrene to $\alpha$-pinene ratio; and the SQT/MT ratio (for suppression).

Line 327. "We suggest that the increased desorption temperatures for the Scots pine experiments 1-3 relative to the $\alpha$-pinene experiment" Do you mean from exps 1 – 3 you see an increase in Tmax and this is explained by increasing contributions of far-nesenes? Or that farnesene concentrations are broadly the same for exps 1 – 3 and have the same (broad) Tmax? Is this why there are only 2 Tmaxes shown on figure 5b, rather than 4?

Line 343. "Some signatures of thermal decomposition are visible as well, but overall this appears to play a minor role, with very small effects on Tmax in most individual ion cases". Please explain what you mean by signatures of thermal decomposition. Do you mean multiple peaks in the thermogram? Please explain how you are treating multiple peaks in thermograms e.g. disregarding or deconvolving multipeak thermograms. Can you be more specific than 'small effects'?

Line 344. "Consequently, the shifts to higher desorption temperatures as observed in the sum thermograms (Fig.5) are essentially seen throughout each respective spec-trum of individual unit mass thermograms, although the contribution of thermal decom-position appears to increase concurrently." I am unsure what this means. It sounds like at all unit masses, you are seeing increasing Tmax as you increase the experiment number.

Line 369. " . . . manifests in the disappearance of SVOC . . . ". Disappearance or reduc-tion?

Line 397. Just OH exposure or oxidation in general if ozonolysis is included? Technical Corrections

Line 44. Missing word. "The volatility of a specific compounds in turn is determined by both its molar mass and functional group composition"

Line 300. There is no Sect. 3.4.

Line 432. Some missing words
* * *

---

## Author Comment (AC1) · 29 Jan 2020

Response to reviewer comments for manuscript: "Composition and volatility of SOA formed from oxidation of real tree emissions compared to  simplified VOC-systems"

Ylisirniö et. al

We thank the two reviewers for their constructive comments regarding the paper. Below we address the specific issues point by point. The reviewer's comments are in black and our answers are in blue. Changes to the Manuscript or Supplement Information are highlighted in red. Line numbers before the red response text refer to line numbers in the modified manuscript.

**Additional changes by authors:**

Due to graphing mistake, all C* values shown in Figures 6-8 were one order of magnitude too high. These figures have been redrawn with correct values. As this change is in all experiments, our conclusions from the figures stay the same.

The language of the text is also slightly adjusted for better readability.

**Reviewer 1:**

**Major comments:**

l. 25: Bisabolenes also have endocyclic double bonds. Their oxidation would not lead to smaller and more volatile compounds. I suggest rephrasing to "[...] is due to exocyclic C=C bond scission [...]"

The text was rephrased as suggested.

l. 62: not only mixtures of VOCs influence the composition of products, but also trace gases like NOx and CO. Furthermore, ammonia and H2SO4 can alter particle phase composition through heterogeneous reactions with organic molecules. It would be good to cite a review at this point (e.g. Ng et al., ACP, 2017, doi:10.5194/acp-17-2103-2017 for the NOx effects).

30

The text was modified as suggested.

l. 82: Was the temperature actually monitored/regulated during the experiments, or does 25∘C just mean

35    that experiments were done in the laboratory? I can imagine that the temperature changes in the flow

tube when UV lamps are switched or when UV exposure is modulated.

Both the temperature in the OFR and in the room were monitored during the experiments. The room

was air conditioned to ~21 °C. Due to the heat from the UV lamps inside OFR, the temperature in the

40    reactor was at ~26 °C after the initial "warm-up" phase. Then the reaction temperature was stable within

+/- 1 °C or less during the experiment. Care was taken to allow sufficient time for the setup to stabilize

before collecting the FIGAERO samples.

45    Section 2.1: A more thorough description of the OFR-experiments is urgently needed. With the current

description, it would be impossible to replicate these experiments. Instead of mentioning already results

in the methods section (l. 84/85, l. 101), the level of information on the setup in the current version is

unsatisfactory. As a reader I like to know for example: How were blank experiments conducted? Was

there a carry-over between different experiments when VOC precursors were changed? Did VOCs /

50    SVOCs come back from the wall when switching on UV lamps? What were the flow rates? How were

the VOCs flushed into the chamber (material of the tubing?)? Were the source- and sampling-lines

heated to avoid condensation of sesquiterpenes?...

We extended the setup description in section 2.1 (including a reference to an earlier study with similar

55    setup (Buchholz et al. 2019)) to the following:

Line 82 onwards:

We conducted experiments with SOA generated from the ozonolysis and photo-oxidation of VOCs by

hydroxyl radicals (OH) in a Potential Aerosol Mass (PAM) OFR (Kang et al., 2007; Lambe et al., 2011)

60    in the absence of seed particles. The experimental setup is similar to that in our previous study

(Buchholz et al., 2019). A schematic of the setup is shown in Figure 1 and all experimental conditions

are listed in Table 1 and Table 2. We provide a very brief description of the experimental setup here and

more detailed information can be found in the SI material (Sect. S2). A flow containing 200 to 400 ppb

of the investigated VOCs was mixed with an $O_3$ containing flow directly before entering the OFR. With

two UV lamps (254 nm), $O_3$ was photolyzed to O (1D) which reacted with water vapor to produce OH radicals. A wide range of OH exposure was achieved by adjusting the voltage of 254-nm UV lamps in the OFR and changing the $O_3$ concentration. Overall, the integrated OH exposure in the OFR ranged from approx. $6.6 \times 10^{10}$ to $2.5 \times 10^{12}$ molec cm$^{-3}$ s across all experiments as calculated according to methods described by Peng et al., (2015, 2016). This range of OH exposure corresponds to 0.5 to 19 equivalent days of atmospheric aging at an OH concentration of $1.5 \times 10^6$ molec cm$^{-3}$ (Palm et al., 2016). In all experiments, the operation temperature of the OFR was 25 or 27 ˚C and relative humidity (RH) was between 40% and 60%. For the Scots pine experiments, VOCs were introduced by flushing purified air through a plant enclosure (Tedlar®) containing a 6-year-old Scots pine sapling. In the α-pinene (Sigma Aldrich, 98 % purity) and sesquiterpene mix (mixture of acyclic/monocyclic sesquiterpenes, Sigma Aldrich) experiments, the VOC's were introduced into a flow of clean $N_2$ by using a diffusion source or a dynamic dilution system (Kari et al., 2018). For the Scots pine experiment 4, the plant was injured by making four 0.5-1 cm$^2$ cuts into the bark of the plant exposing resin pools and thus increasing the VOC emissions.

We also added a more detailed description of the OFR experiments in the supplement material sect. S2:

Line 15 in SI:
SOA was generated by oxidizing different VOCs by OH radicals and $O_3$ in the OFR, in the absence of seed particles. VOCs were introduced by flushing clean air/$N_2$ through different sources. For the Scots pine experiments, VOCs were introduced by flushing purified air through a plant enclosure containing a 6-year-old Scots pine sapling. For α-pinene and sesquiterpenes SOA experiments, VOC vapors were introduced into a flow of dry $N_2$ using a diffusion source or a dynamic dilution system (Kari et al., 2018). In the dynamic dilution system, a set volume of VOCs was continuously injected into a heated $N_2$ flow with a syringe pump. The VOC-containing flow was then mixed with other make-up flows before entering the OFR. To achieve the desired RH, water vapor was introduced by passing a flow of $N_2$ through a Nafion humidifier (Model FC100-80-6MSS, Perma Pure). $O_3$ was generated in an external generator by irradiating a flow of $O_2$ or purified air with a 185-nm UV lamp. The exact ratios of these flows (VOC-, $O_3$, and humidified flow) varied between the different sets of experiments. But all relevant parameters of the mixture entering the OFR were carefully monitored. The mixing ratio of VOCs was continuously measured by a proton transfer reaction time-of-flight mass spectrometer (PTR-MS, PTR-TOF 8000, Ionicon Analytik, Austria) before mixing with $O_3$. To minimize line losses, the combined PFA (~2.5m, 6 mm outer diameter) and PEEK (~1 m, 1/16'' outer diameter) sampling line to the PRT-MS was heated to 60 °C. All other sampling lines were unheated stainless steel or conductive silicon tubing (Tygon®) as this work was focused on the particle phase composition. Overall, 2.5 or 5 L

min$^{-1}$ of mixed flow containing 200 - 400 ppb of VOCs was introduced into the OFR for photooxidation and ozonolysis, with leads to residence times in the OFR ranging from 120 to 300 s (assuming plug flow).

Inside the OFR, O($^1$D) was generated from the photolysis of O$_3$ with 254 nm lamps and reacted with water vapor to form OH radicals. To minimize the impact of heat generated from the 254-nm lamps, we continuously purged the lamps with N$_2$. We varied the OH exposure by adjusting the voltage of the 254-nm lamps inside the OFR and/or the ingoing O$_3$ concentration. The resulting OH exposure ranged from approx. $6.6 \times 10^{10}$ to $2.5 \times 10^{12}$ molec cm$^{-3}$ s in the OFR, using the model calculations described by Peng et al. (2015, 2016) taking the external OH reactivity into account. Assuming an ambient OH concentration of $1.5 \times 10^6$ molec cm$^{-3}$, this range of OH exposure corresponds to 0.5 to 19 equivalent days of atmospheric aging. Before and after each SOA experiment, we always conduced photooxidation cleaning for the OFR for several hours, i.e. flushing the PAM reactor with the same flows as during the experiments but without adding any VOCs. Background particle number concentration decreased to less than 2,000 # cm$^{-3}$, (particle mass < 0.1 µg m$^{-3}$. These values were neglectable in comparison to the $10^6$ – $10^8$ particles # cm$^{-3}$ (and 50 – 500 µg m$^{-3}$) which were formed during the experiments. After the photooxidation cleaning, the VOC concentrations detected with the PTR-MS were within the instrument background. Care was taken to allow sufficient time after the VOC type or concentrations was changed.

l. 107: What does the different "use history" of the PAM chambers mean? That statement leaves the reader questioning whether the history of experiments has an effect on the results?

As laid out in our reply to the previous comment, great care was taken to clean the PAM reactor and all connected lines between experiments, especially when the VOC type was changed.

With the term "usage history" we wanted to refer to the fact that while "PAM1" had been in use for a long time and the UV lamps were getting close to the end of their lifetime, "PAM2" was brand new. This together with other experimental constrains (e.g. sampling flow requirements for instrumentation) led to a different combination of light intensity, residence time, and O$_3$ concentration needed to replicate the OH exposure range and oxidation state of the formed SOA particles.

As the term "usage history" clearly lead to some confusion, we have removed it and rephrased the sentence:

Line 115:

For a detailed description of the mixture see Table S1 and Fig. S1. For those follow-up experiments, a nominally identical OFR was used ("PAM 2"). However, to recreate the same OH exposure and particle composition (as characterised by particle oxidation ratio, see below) a different combination of light intensity, residence time, and O3 concentration was necessary in the follow-up experiments. Thus, the results are presented separately and marked PAM1 or PAM2.

Section 2.2: The level of experimental detail is unsatisfactory. A "suite of instruments" is mentioned in l. 112-115, but the following paragraphs only describe FIGAERO-CIMS and nothing on AMS or SMPS measurements. Concerning AMS, more information is needed in this experimental part, e.g. on the version (HR-AMS or C-ToF), operation mode (V-mode or W-mode) of the AMS, with/without size dependent composition, etc. Furthermore, key information of FIGAERO-CIMS operation mode are also missing: how were blank measurements conducted? Have the authors evaluated whether gas-phase adsorption on the particle filter can be an issue? What was the sample flow rate and the duration of particle collection? Until which maximum desorption temperature the ramp was operated? What was the pressure in the ion molecular region? etc.

More information about the instruments and methods have been added to section 2.2. However, as this study focuses on the particle composition derived from FIGAERO-CIMS measurements, we added the requested additional information about other instruments into the supplement material Sect. S4:

Line 125:
AMS and SMPS were used to continuously monitor the output SOA particle mass and size distribution from the OFR to determine the point when the particle concentrations and distributions had stabilized for a given OFR condition. Then the filter collection for FIGAERO-CIMS was started, so that only steady-state SOA was sampled. More information about these other instruments is given in Sect. S4.

Line 53 in SI:
S4 SOA characterisation
The outflow of the OFR was periodically checked with the PTR-MS to ensure that our assumption of complete consumption of the ingoing VOCs was correct. The same 2.5 m (outer diameter 6 mm) PFA + 1 m (outer diameter 1/16'') PEEK line heated to 60 °C was used for sampling before or after the OFR. When the sampling point was changed sampling lasted at least 30 min to ensure all compounds had reached their final values (especially sesquiterpenes).

The outflow of the OFR was continuously monitored with an AMS and a SMPS. The SMPS was operated with a closed loop sheath flow. The RH and temperature measured in the sheath flow in the instrument was close to the experimental conditions in the OFR.

The AMS was operated in V-mode and although particle size resolved data was collected, only the integrated signal was used for the analysis. The raw data was processed with the SQUIRREL (Version 1.59D) and PIKA toolkits (Version 1.19, Decarlo et al., 2006). As the composition of the "air" in OFR changed depending on the ratio between the $N_2$ and $O_2$ flows introduced into it, a time dependent air beam and $CO_2$ correction was applied. The main purpose of the AMS measurements were to classify the SOA particles by their oxidation state (O:C and H:C ratios, $OS_C$). The improved parameterisation from Canagaratna et al., 2015 was used to derive these values from the data.

We performed a blank measurement with FIGAERO-CIMS before each measurement to make sure the filter was clean of residual compounds and to determine any instrument artefacts. For our samples, the collection time (i.e. the time the filter was exposed to the sample gas and particle phase was short (~2 minutes) and high mass loading (~500 ng – 1350 ng) were collected. Thus, we can assume that generally signals from gas phase adsorption are minor compared to signal stemming from collected particles. However, in the data-analysis we identified a small group of signals (mostly lactic and formic acid) that were suspected to originate from gas phase adsorption or other additional contamination due to the shape of their thermograms. They all exhibited high signals already at the start of the desorption and a smooth increase with increasing temperature. These signals accounted for around 1% of total signal and were excluded from further analysis.

The blank measurements are now discussed in the text:

Line 154:

A blank measurement, meaning a measurement with no particles collected on the filter, was performed before each measurement to make sure the filter was clean of residual compounds and to determine any instrument artefacts. The assure that filter did not contain any particles, collection flow leading to the filter was shut down between actual measurements. These blank measurements were also considered in the data-analysis. The relatively high collected particle mass loading (between 500 and 1350 ng) on the filter ensured that the majority of the signal came from the evaporating SOA particles, and that the instrument background/artefacts were neglectable. The FIGAERO filter was also visually inspected daily and replaced when needed.

205     Figaero data analysis: Why did the authors use tofTools and not Tofware, which offers also a figaeroversion? Concerning peak identification: What are the numbers of allowed elements for C, H and O?

How were multiple possible sum formulas for one exact mass ranked? Were ratios of H/C or O/C and

double bound equivalents used to further constrain sum formula?

210     The reviewer is correct in pointing out that the Tofware software includes a FIGAERO – data-analysis

package. However, the post processing of the FIGAERO-CIMS data required heavy use of custommade analysis scripts, which were done with Matlab. Using tofTools-preprocessing software enabled

keeping the whole analysis inside one programming system.

215     In peak identification, we used upper limits of H/C = 2.5 and O/C = 2 as constraints for possible

compositions. These limits constrain the majority of the signal in each experiment. However, there was

a small number of signals with mass defects that could only be explained with formulas with higher H/C

or O/C value even though such compositions seem unrealistic. These compounds comprised only a

small amount of total signal and thus did not impact the reported average formula. We did not use any

220     DBE limits as previous studies (Kourtchev et al., 2014) have shown that possible DBE value increases

with molecular mass.

    l. 167: Was an experiment conducted to measure wall/sampling line loss rates of sesquiterpenes against

225     monoterpenes? E.g. SQT/MT measurements before and after the OFR, without O3 and UV off?

    We did not conduct such measurements but will consider them in future projects to get a more thorough

characterization of our OFR.

230     We followed the recommended procedure for PTR-MS measurements at the time using a PFA or PEEK

sampling line heated to 60 °C which minimizes any condensation of SQT or MT in the sampling line.

We measured the VOC's 10 cm before the inlet plate of the OFR, which was the closest the PTR-MS

measurement inlet could be connected before addition of $O_3$ to the flow.

235     As we always utilized large OH exposure levels, and there was ppm level of $O_3$ in the OFR, the majority

of SQT and MT should react with oxidants rapidly after entering the OFR. Thus, we would need to

determine the wall loss rates of the products of that oxidation which was far beyond the scope of this

study.

240

L. 171: I disagree with this statement: wall/sampling line losses will be larger for sesquiterpenes than for monoterpenes. Relating this to the observed results, evaluating potential different losses of SQT compared to MT are highly relevant for the conclusions on different SOA yields.

245     This is a good point. To partly counteract this, we minimized the length of tubing between VOC measurement and inlet of the OFR as much as we could. It is possible that some of the SQTs were lost more than MTs in this short 10 cm tubing and losses in the heated PTR-MS sampling tube could also be different for SQTs and MTs. To address this comment, we rephrased this sentence as follows:

250     Line 192:
However, because the VOC measurement was made almost immediately before the inlet of the OFR and formed SOA size distribution was roughly identical between different experiments, we assume that possible sampling line wall losses have minor impact on our SOA yield calculations.

255

Based on Fig. 2: does it need to be reassessed whether a-pinene is the (globally)strongest emitted monoterpene?

We do not believe such a re-assessment is needed. We were looking at the emissions of a single tree at a
260     short time when it was stressed. It is well known that even within one species (here Pinus Sylvestris) there can be many different chemotypes. As an example, there are Pinus Sylvestris trees that emit more Δ-3-carene than a-pinene and those that do not emit this compound at all (Bäck et al. 2012). Also, the time of year plays a major role in the emission patterns (Hakola et al., 2006). This is why global emission inventories are based on a multitude of measurements both directly from branch enclosures
265     and above forest canopies.

Our results highlight that more care must be taken when representing tree emissions in OFR or chamber experiments and that α-pinene is not necessarily representative for all pine forests. These other VOCs (e,g, induced by stressors) may have a significant impact on SOA formation and properties (see also
270     Faiola et al., 2018, 2019)).

l.274-276: this has already been shown by Hall et al.(https://doi.org/10.1021/es303891q)

275    Thank you for pointing this out. We have added a reference to Hall et. al. 2013.

l. 309-316: Is this observation in line with what one would expect from $C^*$ estimates based on the sum formula? (Donahue et al., ACP, 2011). It is surprising that oxygenation outweighs fragmentation of e.g.
280    C20-dimers. Does the VBS distribution look different when the VBS bins are filled based on T_max vs. when they are filled based on the sum-formula-derived volatility?

Sum formula based $C^*$ parametrizations tend to yield very different results between different methods (Mohr et al., 2019) and while comparing them to results made with Tmax based $C^*$ estimates would be
285    very interesting, it would be outside of the scope of this article. Based on volatility estimations shown in Mohr et al., 2019 measured in boreal forest in Finland, composition based volatility estimations seem to predict higher fraction of volatile compounds classified in SVOC class compared to our results.

290

Fig. 6 – SQT mix – mass spec: What are the ELVOCs with molecular mass < 200 Da? Are such compounds formed by thermal fragmentation of larger molecules? If this is the case, a quantitative estimate of the fraction of ion signal that results from thermal fragmentation would help to evaluate the FIGAERO mass spectra.

295

These signals (mass < 200 Da and in the ELVOC region) are C3-C7 compounds with mostly relatively high oxygen content O3-O7. The reviewer is correct when pointing out that they are possibly thermal decomposition products and they comprise about 11% of the total signal.
This is now also pointed out in the text as:

300

Line 408:
A small number of signals in the sesquiterpene mixture results can confidently be categorized as thermal decomposition products, namely the ones that fall in the ELVOC volatility range, but have relatively small molecular masses (< 200 Da). These compounds are C3-C7 compounds with relatively high
305    oxygen content (O3-O7) and comprise about 11% of the total integrated signal.

l. 418: A comparison FIGAERO spectrum between acyclic and mono/bicyclic SQT would be very interesting. Fig. 4 only shows the spectrum of a SQT mix. More fragmentation products would be expected for the acyclic SQTs.

Such comparison of acyclic vs. mono/bicyclic SQT would be indeed very interesting and would help shed light to some questions raised by our results, and potentially provide additional evidence to the presented hypotheses regarding SQT cyclic structures. Unfortunately, we did not have respectively representative SQT mixtures available at the time of our experiments, so that practical time constraints precluded us from conducting the suggested comparative measurements. We hope that we (or others) will follow up with a thorough investigation of FIGAERO spectra from a variety of SQT in the future.

**Minor:**

l. 54 add Yassaa et al., ACP, 2012; doi:10.5194/acp-12-7215-2012.

Citation added.

l. 119: oxygen or OH-groups: consider to be more specific. Would an ether (a molecule with an oxygen) already give a stable cluster with iodide?

The reviewer is correct when pointing this out, as mere existence of oxygen in the chemical formula is not enough for this ionization process. Text is changed to:

Line 132:

… clusters with a neutral molecule *M* which contains hydroxy-, hydroperoxy-, carboxyl- or peroxycarboxyl-groups in their structure.

l. 409: Explaining what fragmentation means should not appear in the summary /conclusions.

The reviewer is correct and following lines have been removed, as they have been already discussed in section 3.2.

Line 447:

"When such acyclic compounds are oxidized, the initial reaction with OH or $O_3$ already leads to fragmentation (i.e. reaction products with much shorter C chain), whereas cyclic compounds, such as many monoterpenes, (including α-pinene and β-phellandrene) but also bicyclic sesquiterpenes (such as β-caryophyllene), generally go through more steps of oxidation before the onset of significant fragmentation is observed."

**Technical:**

l. 52 replace a-pinene with α-pinene.

Fixed the text.

l. 62, l. 269 replace Mcfiggans with McFiggans.

Fixed the references.

Fig. 2: structural formula of farnesene expliciltly shows one hydrogen. Either draw all hydrogens or none (preferably none),

Figure modified so that no hydrogens are shown in the structure.

**Reviewer 2:**

**Major comments:**

Line 26. "SOA particles from the oxidation of Scots pine emission had similar or lower volatility than SOA particles formed from either of single precursor." Does the single precursors refer to the SQT
360    mixture as well as the alpha pinene? If so that might need rephrasing. This is also true for the title.

Line 28:
Text rephrased as "…SOA particles formed from either single precursor or simple mixture of VOCs."

365    Title is rephrased as "Composition and volatility of SOA formed from oxidation of real tree emissions compared to simplified VOC-systems".

Line 30. You state "These results emphasize that simple increase or decrease of relative monoterpene
370    and sesquiterpene emissions should not be used as indicator of SOA particle volatility" but on line 27 you state "Applying physical stress to the Scotspine plants increased monoterpene emissions, which further decreased SOA particle volatility and increased SOA mass yield." Can you clarify or rephrase the statement online 30 based on the statement from line 27.

375    Statement is meant to highlight that change of total monoterpene or sesquiterpene emissions without stating the structures of these compounds should not be used as indicator for SOA properties. Text has been modified to emphasize this.

380    Line 87. Could you please provide a brief description on the dynamic dilution system you use to mix your VOCs.

The dynamic dilution system has been extensively described in earlier publication by Kari et al. 2018 (Int. Journ. of Mass Spectrometry). The reference is added to the setup description.
385    Briefly, we use a syringe pump to continuously inject a VOC (or VOC mixture) into a nitrogen flow heated to 35 – 60 °C. A portion of this flow was than mixed with the other flows going to the OFR.

Section 2.2.1/S3 There is some missing information that would be useful. Can you please state the resolution of the ToF-CIMS. A description of your FIGAERO back-grounding methodology and how you account for backgrounds in your analysis is missing. Also a description of your gas and particle sample lines to the ToF-CIMS (material, length, residence time etc).

As Reviewer #1 also requested more detailed information about the setup in general, we extended the methods section and added the more specific details into the supplement material. Please refer to our response to Reviewer #1 above.

Line 169. Although you did not correct for wall losses, can you make an estimation or assessment of their importance? This may be important if the yields from this paper are compared with other experiments.

Directly determining the wall losses of just the main reaction products of α-pinene would be a project on its own. Instead we inspected the model calculations by Palm et al., 2016. They determined the fate of LVOCs in a PAM OFR in relation to the OH exposure. The residence time in their system was comparable to ours, but they were focusing on oxidizing ambient air samples. LVOCs are used as a proxy for condensable vapors/ vapors contributing to SOA formation and growth in the OFR. They estimate that in the absence of any particles, 30% of the LVOC vapors would condense on the OFR walls and the rest would be further oxidized or leave the OFR and potentially condense on tubing surfaces etc. Fig. 5 in Palm et al. (2016) can be used for a qualitative assessment of wall losses once SOA particles are formed in our experiments. Quickly, the "high Condensational Sink" conditions are met, meaning that for our OH exposure range < 15% of LVOC vapours are lost to the OFR walls. Thus, the SOA yields would be underestimated by a similar amount due to LVOC wall losses.

The dynamics of the particle formation will play an important role as we did not use seed particles in these experiments. A more detailed model description of the flow dynamics and chemistry in the OFR would be needed to understand the impact of the different VOC precursors and the different flow setup on the wall losses.

These potential differences in the losses of particles and gaseous compounds is one of the reasons we present results from PAM 1 and PAM 2 setup separately. The different residence times and OH distribution may lead to different losses.

Line 186. Can you provide some more detail on the scrubber / filter for the compressed air.

We used a custom build air purifying unit consisting of a sequence of scrubbers with active charcoal, potassium permanganate and filters to dry the compressed air and clean out any VOCs.

Line 190. Can you provide more information on the LED lamps.

The LED lamps were chosen to create a Photosynthetically Active Radiation (PAR) value similar to that experienced by plants in the Finnish environment. This is now also stated in the text.

Line 210. Is the stress response induced by cutting the sapling reduced or altered because it is already stressed by the infestation? Are there any further considerations required as to how these two stresses alter VOC emission regarding the conclusions you can draw?

The effect of cutting the bark of Scots Pine seedlings is comparable to the damage induced by infestation with bark borers. The main impact on the plant is the exposure of resin pools in the stem (Faiola et al. 2018, Kari et al 2019). As the content of the resin pools is dominated by the longer-term storage and not the immediate production of the infested plant, it can be assumed that the interaction of these two stressors has a very small impact on the emission profiles.

Line 214. Is it shown that MT concentrations were higher? Experiment 2 shows the highest [MT] concentration (table 1). Do you mean the transition from experiment 3 to experiment 4 specifically?

Monoterpene emissions increased from to cutting the sapling by about doubling the monoterpene concentration once the concentrations had stabilized compared to pre-cutting concentrations. This has been clarified in the text as:

Line 237:
"The wounds exposed plants resin pools in the stem and increased the measured monoterpene concentrations compared to pre-cutting concentrations by roughly doubling them."

Line 227. "...are about 30% larger, even though SQT/MT was substantially smaller than in experiments 1-3." I find the words 'even though' confusing, can you explain why these findings are unexpected?

460     Thank you for pointing this out. The surprise in question stems from results reported earlier by Faiola et. al. 2018 who measured positive correlation with SOA yield and SQT/MT ratio, when they measured SOA formed from VOC's emitted by Scots pine seedlings in similar flow tube experiments as ours. The text has been modified to point out this detail and its then discussed in more detail later in the text.

465     Line 251:

Earlier study by (Faiola et al., 2018) reported a positive correlation with SOA yield and SQT/MT ratio of VOC's measured from Scots pine seedlings in a flow tube experiment similar to our study.

470     Line 241. "We conclude that the increase in SOA yield in the Scots pine experiment 4, compared to the Scots pine experiments 1-3, is likely due to the large relative increase in emitted monoterpenes" I think more clarification is needed regarding this conclusion. Can you provide a concise explanation of which factors regarding the MT are important for the increased SOA yield in experiment 4, such as absolute [MT] increase induced by cutting; MT composition change induced by cutting e.g.β-phellandrene to α-

475     pinene ratio; and the SQT/MT ratio (for suppression).

The absolute MT concentration first increased by several orders of magnitude and then stabilized to about 2x the MT concentration before the cutting, as stated in previous comment. Even though the absolute MT concentration in Scots pine experiment 4 is not substantially higher than in Scots pine

480     experiment 3 (127 ppb in exp. 3 vs. 150 ppb in exp. 4), the amount of SQT's are noticeably lower (151 ppb in exp. 3 vs 66 ppb in exp. 4). As we discuss at the end of section 3.2, what most strikingly influences the amount of SOA production is the structure of each VOC compound, in general regardless of its classification as MT or SQT. We believe that concentration of all MTs increased as a result of the tree being cut, but unfortunately there is no GC-data available immediately before and after the cut.

485

Line 327. "We suggest that the increased desorption temperatures for the Scots pine experiments 1-3 relative to the α-pinene experiment" Do you mean from exps 1 – 3 you see an increase in Tmax and this is explained by increasing contributions of farnesenes? Or that farnesene concentrations are broadly the

490     same for exps 1 – 3 and have the same (broad) Tmax? Is this why there are only 2 Tmaxes shown on figure 5b, rather than 4?

Scots pine experiments 1-3 were broadly similar to each other in terms of desorption behavior and VOC concentrations, which is indeed why they are compared to Scots pine experiment 4 as a group and why only 2 Tmaxes are shown in Figure 5b). As the Tmax values of Scots pine experiments 1-3 are also quite close to each other, only Tmax values from Scots pine exp. 1 and Scots pine exp.4 are shown in the Fig 5b). This is now clarified in the Figure 5 caption text.

Figure 5:

$T_{max}$ values of the sum thermograms are shown with dashed lines. In panel b) Tmax values of only Scots pine experiment 1 and Scots pine experiment 4 are shown for clarity.

Line 343. "Some signatures of thermal decomposition are visible as well, but overall this appears to play a minor role, with very small effects on Tmax in most individual ion cases". Please explain what you mean by signatures of thermal decomposition. Do you mean multiple peaks in the thermogram? Please explain how you are treating multiple peaks in thermograms e.g. disregarding or deconvolving multipeak thermograms. Can you be more specific than 'small effects'?

By "signatures of thermal decomposition" we indeed meant either multiple peaks or pronounced tailing in thermogram of a single fitted composition. This kind of behaviour was only observed in a handful of signals while the total amount of fitted peaks totals more than 600 compounds. Therefore, Tmax of these multiply peaking thermograms is assigned to the highest peak, which is usually the first mode of the signal and the second mode is disregarded. The second mode could be deconvoluted from the first mode in clear multiple-peak cases, but more often than not such deconvolution, e.g. by fitting multiple peaks, appeared not feasible. The main reason for that is that the applicable peak *shapes* generally remain ambiguous (Lopez-Hilfiker et al., ACP, 2015; Stark et al., EST, 2017; Schobesberger et al., ACP, 2018; Buchholz et al., ACPD, 2019). Therefore, we decided not to involve such deconvolution in this analysis but note that by neglecting secondary thermogram "modes", the use of Tmax values tends to somewhat underestimate volatility (e.g. as expressed by derived saturation vapour concentrations).

The sentence "very small effects on Tmax in most individual ion cases" is poor choice of wording and should read "very small effects on the thermogram in most individual cases" as tailing of additional peaks in the thermogram would hardly affect the allocation of the Tmax value, irrespective of possibly first deconvoluting the thermogram. This is now corrected to the text at line 371.

Line 344. "Consequently, the shifts to higher desorption temperatures as observed in the sum thermograms (Fig.5) are essentially seen throughout each respective spectrum of individual unit mass thermograms, although the contribution of thermal decomposition appears to increase concurrently." I am unsure what this means. It sounds like at all unit masses, you are seeing increasing Tmax as you increase the experiment number.

The sentence clearly needs clarification. It has been rephrased as:

Line 373:
The resistance to thermal desorption at each unit mass appears to increase with increasing strength of oxidation in the α-pinene experiments, as is observed in the sum thermograms (Fig. 5), while the contribution of thermal decomposition appears to increase concurrently. The Scots pine experiments show similar effects between Scots pine experiments 1-4, but the change is not as pronounced as with α-pinene experiments and cannot be as clearly attributed to a single factor such as oxidative strength.

Line 369. "...manifests in the disappearance of SVOC...". Disappearance or reduction?

"Reduction" is the more appropriate term. Term changed in the text at line 404.

Line 397. Just OH exposure or oxidation in general if ozonolysis is included?

The text is rephrased to "oxidative strength" at line 436.

**Technical Corrections**

Line 44. Missing word. "The volatility of a specific compounds in turn is determined by both its molar mass and functional group composition"

Fixed the sentence.

Line 300. There is no Sect. 3.4.

Corrected the section numbering.

Line 432. Some missing words.

565 Corrected the text to past sentence.

Canagaratna, M. R., Jimenez, J. L., Kroll, J. H., Chen, Q., Kessler, S. H., Massoli, P., Hildebrandt Ruiz, L., Fortner, E., Williams, L. R., Wilson, K. R., Surratt, J. D., Donahue, N. M., Jayne, J. T. and Worsnop, D. R.: Elemental ratio measurements of organic compounds using aerosol mass spectrometry : characterization ,

570 improved calibration , and implications, Atmos. Chem. Phys., 15, 253–272, doi:10.5194/acp-15-253-2015, 2015.

Decarlo, P. F., Kimmel, J. R., Trimborn, A., Northway, M. J., Jayne, J. T., Aiken, A. C., Gonin, M., Fuhrer, K., Horvath, T., Docherty, K. S., Worsnop, D. R. and Jimenez, J. L.: Aerosol Mass Spectrometer, , 78(24), 8281–8289, doi:10.1021/ac061249n, 2006.

575 Faiola, C. L., Buchholz, A., Kari, E., Yli-Pirilä, P., Holopainen, J. K., Kivimäenpää, M., Miettinen, P., Worsnop, D. R., Lehtinen, K. E. J., Guenther, A. B. and Virtanen, A.: Terpene Composition Complexity Controls Secondary Organic Aerosol Yields from Scots Pine Volatile Emissions, Sci. Rep., 8(1), 1–13, doi:10.1038/s41598-018-21045-1, 2018.

Faiola, C. L., Pullinen, I., Buchholz, A., Khalaj, F., Ylisirnio, A., Kari, E., Schobesberger, S., Yli-juuti, T.,

580 Miettinen, P., Holopainen, J. K. and Kivima, M.: Secondary Organic Aerosol Formation from Healthy and Aphid- Stressed Scots Pine Emissions, , doi:10.1021/acsearthspacechem.9b00118, 2019.

Hakola, H., Tarvainen, V., Bäck, J., Ranta, H., Bonn, B., Rinne, J. and Kulmala, M.: Seasonal variation of mono- and sesquiterpene emission rates of Scots pine, Biogeosciences, 3, 93–101, 2006.

Kourtchev, I., Fuller, S. J., Giorio, C., Healy, R. M., Wilson, E., Connor, I. O., Wenger, J. C., Mcleod, M.,

585 Aalto, J., Ruuskanen, T. M., Maenhaut, W., Jones, R., Venables, D. S., Sodeau, J. R., Kulmala, M. and Kalberer, M.: Molecular composition of biogenic secondary organic aerosols using ultrahigh-resolution mass spectrometry : comparing laboratory and field studies, Atmos. Chem. Phys., 14, 2155–2167, doi:10.5194/acp-14-2155-2014, 2014.

Mohr, C., Thornton, J. A., Heitto, A., Lopez-hil, F. D., Lutz, A., Riipinen, I., Hong, J., Donahue, N. M.,

590 Hallquist, M., Petäjä, T., Kulmala, M. and Yli-juuti, T.: Molecular identification of organic vapors driving atmospheric nanoparticle growth, Nat. Commun., 10(4442), 1–7, doi:10.1038/s41467-019-12473-2, 2019.

Palm, B. B., Campuzano-jost, P., Ortega, A. M., Day, D. A., Kaser, L., Jud, W., Karl, T., Hansel, A., Hunter,

J. F., Cross, E. S., Kroll, J. H., Peng, Z., Brune, W. H. and Jimenez, J. L.: In situ secondary organic aerosol formation from ambient pine forest air using an oxidation flow reactor, , 2943–2970, doi:10.5194/acp-16-2943-2016, 2016.

595

---

## Author Response (AR2)

**Response to Editors comments to the Author in "Arttu Ylisirniö *et. al.*,: Composition and volatility of SOA formed from oxidation of real tree emissions compared to simplified VOC-systems".**

Editors comments are shown on black text and our response to them is on blue.
* * *
l. 64: NOx (subscript x), corrected

l. 155: replace 'The assure' by 'To assure', corrected

l. 259: replace a-pinene by alpha-pinene (i.e. use Greek symbol), corrected

l. 288: 'creates' seems redundant, removed "creates" from the text

l. 353: replace 'is' by 'are', corrected

l. 418: remove '#' in first parentheses or add it in second to be consistent, added "#" to the second parentheses

l. 445: 'the relatively high (> 5) amount of oxygen atoms in a major fraction of the SOA compositions' should be reworded, e.g., 'the relatively high (> 5) number of oxygen atoms in a major fraction of the products comprising SOA', corrected as suggested

l. 455: 'O:C/OSc values' reads as if you refer to the ratio of the O:C to OSc values. Please replace accordingly. "O:C/OSc" corrected to "O:C and OSc"

Figure 3, caption: (i) l. 653/658: the word 'panel' can be removed in both instances; (ii) l. 654: 'In these cases, and error bars are omitted for clarity' seems incomplete or grammatically wrong. Removed "and" from the sentence. The text reads now: "In these cases, error bars are omitted for clarity".

Figure 6, caption: (i) l. 674: 'max' should be subscript; (ii) l. 677: should 'in the volatility classes VBS' read 'in the VBS volatility classes (or 'bins')'? Corrected the subscript and removed word "VBS" from the end of the sentence.

Figure 7, caption: l. 683: replace 'show' by 'shows', corrected as suggested

Figure 8: (i) In all panels, Scotspine' should be replaced by 'Scots pine'. Figure corrected

Data availability:
Please follow the journal policy for data availability.
https://www.atmospheric-chemistry-and-physics.net/for_authors/manuscript_preparation.html
Make data accessible or - if the data are not publicly accessible - add a detailed explanation of why this is the case (e.g. applicable laws, university and research institution policies, funder terms, privacy, intellectual property and licensing agreements, and the ethical context of the research).

Added additional contact information for data request.